# Predicting the Emergence of Induction Heads
# in Language Model Pretraining

**Tatsuya Aoyama** [1]  **Ethan Gotlieb Wilcox** [1]  **Nathan Schneider** [1]

## Abstract

Specialized attention heads dubbed *induction heads* (IHs) have been argued to underlie the remarkable in-context learning capabilities of modern language models; yet, a precise characterization of their emergence, especially in the context of language modeling, remains wanting. In this study, we investigate the relationship between statistical properties of the training data and IH formation in both natural and synthetic training data settings. We show that: (1) a simple equation combining batch size and context size predicts the point at which IHs form and that this emergence point is agnostic to model size; (2) surface bigram repetition frequency and reliability strongly affect the formation of IHs, and we find an effective decision boundary in terms of these two values; (3) local dependency with high bigram repetition frequency and reliability is sufficient for IH formation, but categoriality and the shape of the marginal distribution appear to modulate IH formation near the decision boundary.

## 1. Introduction

The practical utility of modern large language models (LLMs) depends heavily on their ability to perform in-context learning (ICL), broadly construed as performing a task based on the input provided at inference time. Various accounts have been provided to explain the internal workings of this capability, and among them are studies that find certain attention heads affecting language models (LMs)' ICL capabilities (Olsson et al., 2022; Edelman et al., 2024; Reddy, 2024; Yin & Steinhardt, 2025). Olsson et al. (2022) maintain that specialized heads that emerge abruptly during pretraining, dubbed induction heads (IHs), are primarily responsible for the emergent ICL capabili-

ties LMs exhibit. Edelman et al. (2024) mathematically show ICL is equivalent to a type of Bayesian inference, and that IHs are indispensable in making this inference. The emergence of such heads during pretraining is sometimes referred to as phase change or phase transition (Chen et al., 2024; Aoyama & Wilcox, 2025). The broader phenomenon of phase transition is not limited to the text modality, and has been observed in vision models (Okawa et al., 2023; Park et al., 2024).

It has been shown that IHs form in LMs very early in pretraining, regardless of the model size, as long as the model has multiple heads and at least 2 layers (Olsson et al., 2022). While the behavior and mechanism of IHs are relatively well-understood (Elhage et al., 2021; Olsson et al., 2022; Edelman et al., 2024), it is not clear *when* and *why* such heads form naturally when trained on language data. In other words, we ask: when exactly do IHs emerge in LMs, and what properties of natural language give rise to IH formation, potentially by incentivizing the copying behavior in transformer LMs? Given that the emergence of IHs has downstream behavioral consequences, precisely characterizing their emergence point has practical implications for LM pretraining, such as dataset selection and training efficiency.

In this study, through a series of experiments using both natural and synthetic data, we draw the following conclusions: (1) a simple equation combining batch size and context size can predict the point at which IHs form and that this emergence point is agnostic to model size (Section 6.1); (2) the surface bigram repetition frequency and reliability strongly affect the formation of IHs, and a decision boundary can be described in terms of these two values (Section 7); (3) local dependency with high bigram repetition frequency and reliability is sufficient for IH formation, but categoriality and the shape of the marginal distribution appear to modulate IH formation near the decision boundary (Section 8). Finally, we also show that the emergence points relate non-monotonically to final validation loss, suggesting their relevance to broader training dynamics (Section 9).[1][2]

---

[1]Department of Linguistics, Georgetown University. Correspondence to: Tatsuya Aoyama <ta571@georgetown.edu>.

*Proceedings of the 43rd International Conference on Machine Learning*, Seoul, South Korea. PMLR 306, 2026. Copyright 2026 by the author(s).

[1]Code available at https://github.com/t-aoyam/predict-ih/.

[2]We will use emergence, phase change, and phase transition interchangeably in this paper.

## 2. Relevant Work

Factors that affect phase transition, particularly the formation of IHs, are not fully understood. IHs are often associated with LMs' ICL capabilities.[3] Chan et al. (2022) study the data properties that lead to different learning outcomes in few-shot image classification, where few-shot examples are image-label pairs. They find that (1) burstiness (similar things appearing in clusters), (2) within-class variation, and (3) dynamic class membership all promote ICL and demote in-weight learning (IWL). Interestingly, making the marginal distribution of the labels Zipfian was the only variable that led to high ICL and IWL simultaneously.

Edelman et al. (2024) propose a synthetic task dubbed ICL Markov Chain (ICL-MC) using a Markov Process involving 2-8 symbols. Taking a Bayesian approach to ICL (Xie et al., 2022), Edelman et al. (2024) randomly initialize the transition matrix at the beginning of each epoch, thereby making it impossible to learn the underlying distribution of the symbols. They find that the models sequentially go through phases where their predictions most closely match the uniform distribution, then in-context unigram counts, and lastly in-context bigram counts. They show that by forming IHs that attend to all of the bigram continuations of the current token in the preceding context, models achieve the Bayes-optimal bigram solution.

ICL is often studied as few-shot learning (e.g., Chan et al., 2022; Singh et al., 2024) with input-label (e.g., image-label) pairs, or with a synthetic setting that necessitates ICL as opposed to IWL (e.g., Edelman et al., 2024; Park et al., 2025). Park et al. (2025) propose a novel synthetic sequence modeling task using a mixture of Markov chains; however, in LM pretraining, it is unlikely that each batch comes from a completely different distribution of tokens. As such, it is yet to be clear why LMs, which are trained on natural language through next token prediction, form IHs (and ICL capabilities), and what properties of natural language promote such learning dynamics. Furthermore, the point at which IH formation occurs is not fully understood. Some report a narrow range (1B–3B pretraining tokens; e.g., Olsson et al., 2022) while others find a much wider range (64M–2B pretraining tokens; Aoyama & Wilcox, 2025). Zucchet et al. (2025) study the emergence point; however, they focus on a general framework called sparse attention using a synthetic associative recall task. We focus specifically on IHs in the naturalistic language modeling task.

To fill these gaps, in this study, we aim to find (1) the point at which IHs emerge, and (2) precise data properties that promote the formation of IHs.

---

[3]Olsson et al. (2022) claim that IHs are primarily responsible for ICL, whereas Yin & Steinhardt (2025) find that a different set of heads, dubbed *function vector heads*, are performing few-shot-learning-style ICL.

## 3. Methods

### 3.1. Metrics

IHs are defined by the copying behavior. If the model has seen an $\langle A, B \rangle$ sequence *in-context* and the current token is A, then an IH is a head that promotes the prediction of B as the next token, completing the $\langle A, B, \ldots, A, B \rangle$ sequence.

**Prefix-matching Score.** Following Olsson et al. (2022), we quantify this behavior using prefix-matching score (PS). Given a random sequence of tokens $\mathbf{x}$ repeated twice, PS of a head $h$ at layer $l$ is its average attention from the source token $x_i$ to the next token of its previous occurrence:

$$\frac{1}{|\mathbf{x}|} \sum_{i=|\mathbf{x}|+1}^{2|\mathbf{x}|} \alpha^{(h,l)}(x_i, x_{i-(|\mathbf{x}|-1)}) \tag{1}$$

As for the size of the sequence $\mathbf{x}$, TransformerLens library (Nanda & Bloom, 2022) adopts $|\mathbf{x}| = 50$, which we also do; however, when analyzing models with smaller context sizes, we adjust $|\mathbf{x}|$ accordingly: $|\mathbf{x}| = \min(\frac{|\text{context}|}{2}, 50)$.

**Logit Attribution.** Similarly, Olsson et al. (2022) define a metric called logit attribution (LA). As opposed to PS, which measures a given head's attention to the token of interest, LA measures the actual contribution of a given head's output to the model's final logit of the token of interest via the residual stream. The raw logit contribution $C \in \mathbb{R}^{|\mathcal{V}|}$ of an output $x^{h,l}$ from a given head $h$ at layer $l$ is computed by passing it through the linear layer of the attention block $W_O$ and the final unembedding layer $U$:

$$(x^{h,l} W_O) U \tag{2}$$

Following Olsson et al. (2022), we then center and normalize the positive logit contribution to all tokens in the test sequence $\mathbf{x}$ and compute the ratio of the logit contribution to the target token to the rest of the tokens in the sample. Note that since all models used in this study (GPT2, Pythia) employ pre-layer normalization, there exists a direct path from a given head's output to the final logit calculation. LA captures each head's contribution through this path. However, it is important to note that the rest of the contribution (e.g., through subsequent layers) is not captured in LA.

**Associative Recall.** Lastly, we also measure how often a model predicts B given $\langle A, B, \ldots, A \rangle$, a metric referred to as associative recall (AR). It can be measured by accuracy:

$$\frac{1}{|\mathbf{x}|} \sum_{i=|\mathbf{x}|+1}^{2|\mathbf{x}|} \mathbb{1}\{f(\mathbf{x}_{<i}) = x_i\} \tag{3}$$

and by the mean rank of the target token B:

$$\frac{1}{|\mathbf{x}|} \sum_{i=|\mathbf{x}|+1}^{2|\mathbf{x}|} \text{rank}(x_i; f(\mathbf{x}_{<i})) \tag{4}$$

As opposed to the previous two metrics, which are head-level, these metrics are model-level. As reported later, the abrupt improvement in AR always follows that in PS and LA, in line with the findings from Reddy (2024). Also, given a high correlation between PS and LA, we report the analyses of PS in the main body of this paper.

### 3.2. Models and Checkpoints

At least 2 attention layers have been shown to be necessary for the model to perform the induction task (Olsson et al., 2022; Reddy, 2024; Ekbote et al., 2025) unless the model's hidden dimension is increased exponentially (Sanford et al., 2024). In fact, models with 2 layers have recently been shown to be sufficient to approximate any-order Markov chain (Ekbote et al., 2025). Given these findings, we use a 50M-parameter GPT2 (Radford et al., 2019) with 2 layers and 8 attention heads per layer with the hidden dimension of 768 for all experiments. For the first experiment (Section 5), to study the effect of model size, we also train larger models with 125M parameters and 350M parameters and include models up to 7B parameters for inference. See Table 2 for the details on model architecture and training setup.

For experiments with natural language (Section 5), we adopt a pretrained GPT2 tokenizer with a vocabulary size of 50,257, unless otherwise specified. All models analyzed in this study were trained from scratch for 1B pretraining tokens. We save intermediate checkpoints at 250K and 500K tokens, [1M, 10M] tokens at 1M increments, [10M, 100M] tokens at 10M increments, and [100M, 1B] tokens at 100M increments, resulting in 30 checkpoints per model. We additionally use pretrained Pythia models (Biderman et al., 2023) for follow-up analyses. Note that Pythia models were deemed the only available model family with early enough checkpoints available to study the emergence of IHs, which is known to happen very early in pretraining (64M-3B pretraining tokens; Olsson et al., 2022; Aoyama & Wilcox, 2025). For example, OLMo models provide checkpoints for every 1000 steps of pretraining (Groeneveld et al., 2024), meaning that the earliest model checkpoint is already trained on 4B tokens, likely after the phase transition of interest.

For experiments with synthetic language (Sections 7 and 8), we adopt a vocabulary size of 10,000. We do not use a tokenizer as we only work with token IDs. Note that the change in vocabulary size will likely affect the emergence points (Singh et al., 2024; Zucchet et al., 2025); however, this should not change the results of our analyses in these experiments, as we only focus on *whether* IHs emerge or not, and not on *when* they do, in Section 7 and Section 8.

### 3.3. Data

For natural texts (Section 5), unless otherwise specified, we use the English subcorpus from the Common Crawl

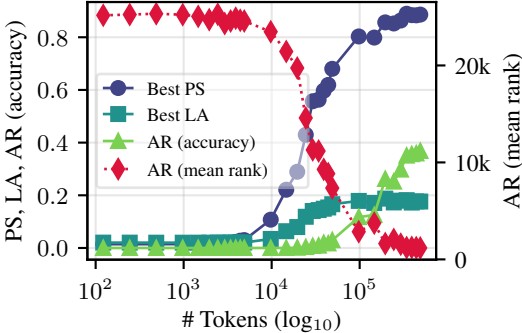

*Figure 1.* Developmental trajectories of PS, LA, AR accuracy, and AR mean rank. The first three metrics are plotted in the scale on the left $y$-axis, and the last metric in the scale on the right $y$-axis.

Corpus (CC100; Conneau et al., 2020; Wenzek et al., 2020). We create a sample of 1B tokens from this corpus, using the pretrained GPT2 tokenizer. As mentioned earlier, all models are trained for 1 epoch on this sample, for a total of 1B tokens. For semi-natural data (Section 7) and synthetic data (Section 8), we use a token-to-token transition matrix, and the details are provided in each section.

## 4. Metric Selection

Before diving into the main experiments, as briefly mentioned earlier, we show how PS, LA, and AR develop during pretraining. Figure 1 shows a sample model (GPT2 125M) trained with batch size of 16 and context size of 128. Clearly, all four metrics go through an abrupt change at around the same time. Notably, AR accuracy seems to increase slightly later compared to the other three metrics. It is important to recall that the best PS and best LA reflect head-level changes in the strategy a model is employing, and that AR (mean rank) measures the model's change in the direction of predicting the target token B, even when it is not the most probable token. On the other hand, AR (accuracy) only improves when the target token B is the most probable token; in other words, the model's improvement from ranking the token B as the least probable (rank 50,257) to second most probable (rank 2) is not captured in this metric. Given these observations, for simplicity and readability, we only report PS in the main body of this paper, while replicating similar analyses with other metrics in Appendix K.

## 5. Experiment 1: Natural Data

### 5.1. Methods

As briefly introduced earlier, Aoyama & Wilcox (2025) find that training an LM with different batch sizes results in different phase transition points. In this experiment, we change context size and batch size to investigate their effect

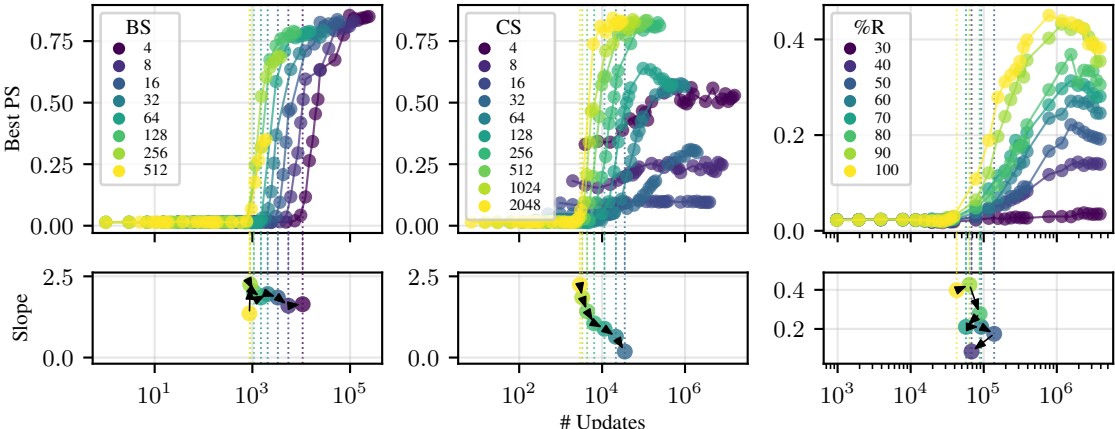

*Figure 2.* Developmental trajectories of PS of LMs with various batch sizes (left), context sizes (center), and repetitions (right) over the course of 1B tokens of pretraining, plotted against the number of updates (a) and the number of tokens (b). BS, CS, %R stand for batch size, context size, and the proportion of chunks with natural bigram repetitions, respectively. The bottom figure plots the identified "emergence point" on the $x$-axis and the slope of the corresponding segment of the curve on the $y$-axis. Configurations that did not lead to the emergence of IHs were excluded from the bottom plot (i.e. context size 4, 8, 16, and %R 30).

on the formation of IHs. Specifically, we experiment with log-spaced batch and context sizes ranging from 4 to 512 and 4 to 2048, respectively. Since grid search is expensive, we fix the batch size at 16 while changing the context size, and fix the context size at 1024 while changing the batch size. We additionally test the effect of bigram repetitions alone by selecting subsets of the pretraining data.

### 5.2. Results

To describe the effect of each of the 3 variables, we identify 2 types of effect: *shifting* and *slanting*. *Shifting* is the change in the point at which IHs emerge, which we identify by fitting a piecewise linear function (PWLF) to the PS curve and taking the first knot as the emergence point. *Slanting* is the change in the slope of the fitted PWLF after the emergence (Section C.1). Figure 2 (bottom) plots the emergence points on the $x$-axis and the slope on the $y$-axis; hence, a movement along the $x$-axis corresponds to the *shifting* effect, and movement along the $y$-axis corresponds to the *slanting* effect.

**Batch Size.** In Figure 2 (top left), we find (1) the larger the batch size, the lower the eventual PS; in other words, a larger batch size leads to weaker IHs at the end of the pretraining, and (2) the smaller the batch size, the later the "spike" in PS; in other words, training an LM with a smaller batch size results in later emergence of IHs, as measured by the number of updates. The bottom plot confirms this shifting effect: decreasing the batch size has little and mixed effects on the slope (i.e., no slanting; $\rho = 0.29, p = 0.49$), but a monotonically positive effect on the emergence point ($\rho = -1.0, p < 0.001$).

**Context Size.** In Figure 2 (center), we find that (1) the smaller the context size, the later the emergence (shifting), and (2) the smaller the context size, the flatter the slope (slanting), and the extreme case (context size $\leq 16$) is a flat line, or the complete suppression of IHs (see Appendix A on how we determine "random" attention). Both of these effects are confirmed in the bottom plot, where the emergence point is monotonically increasing ($\rho = -1.0, p < 0.001$) and the slope is monotonically decreasing ($\rho = 1.0, p < 0.001$) as we reduce the context size. For the shifting effect in (1), since this effect was observed both for batch and context size, we suspect that this could be attributed to the number of tokens the model is exposed to at each update. For the slanting effect in (2), we suspect that, in natural texts, a larger context size will naturally contain more occurrences of $\langle A, B, \ldots, A, B \rangle$ patterns, which may have a threshold below which IHs will not form.

Note that we observe an inverse shifting effect when plotting against the number of pretraining *tokens* in Figure 8 in Appendix B, thereby ruling out the possibility that the observed shifting effect is an artifact of each point on the $x$-axis representing a different number of pretraining *tokens*.

**Repetition.** The number of bigram repetitions increases as context size grows (see Figure 9 in Appendix E), which we hypothesize to cause the slanting effect of the context size. To tease apart these two phenomena, we manipulate the occurrence rate of bigram repetitions within each chunk, while controlling for the batch size and context size. See Appendix E for more details on how we manipulate the repetition rate while controlling for the context size. In Figure 2 (right), (1) the higher the proportion of chunks with bigram repetitions, the higher the best PS a given model

achieves, and (2) we only observe the slanting effect and no shifting effect. This is again confirmed in the bottom plot, where the effect on the emergence point is insignificant ($\rho = -0.64, p = 0.12$), but the effect on the slope is significant ($\rho = 0.93, p = 0.002$). This confirms the observation earlier that the shifting is due to the number of tokens an LM sees per update, and slanting is due to the rate with which an LM encounters repeated bigrams.

## 6. Predictability and Robustness

### 6.1. Predictive Law

We have seen that context size and batch size affect the phase transition point. Here, we ask: can we predict the phase transition point only using the training configuration as variables (i.e., before we train the model)? We start with a full regression model that predicts the number of updates at which IHs emerge based on batch size, context size, and model size. However, as the model size turns out to be the only *non*-significant predictor (see Appendix C for details), we proceed with a regression model without model size:

$$U_{\text{PT}} = e^{\alpha} B^{\beta} C^{\gamma}, \qquad (5)$$

where $B$ and $C$ represent **B**atch size, **C**ontext size, respectively, and $\beta$, and $\gamma$ are their corresponding coefficients. $e^{\alpha}$ serves as an intercept, as we will see below. Following Kaplan et al. (2020), we estimate the parameters $\alpha$, $\beta$, and $\gamma$ using a simple ordinary least squares regression in log space and obtain $\alpha = 13.26$, $\beta = -0.37$, and $\gamma = -0.62$ (see Table 4 for all test statistics). For notational simplicity, let $e^{\alpha} = T$. Plugging these parameters back into Equation (5), we obtain:

$$U_{\text{PT}} = \frac{T}{B^{0.37}C^{0.62}}$$
$$T = U_{\text{PT}} B^{0.37} C^{0.62} \qquad (6)$$

The key intuition behind Equation (6) is that the model-agnostic constant $T$ is a function of the *quantity* of training, or the number of updates $U_{\text{PT}}$, at which phase transition occurs. At the same time, the *quality* of each update matters, and it correlates with the number of tokens the model sees at each update, which is a function of context and batch sizes $B^{0.37}C^{0.62}$. We call the generalized form of the right hand side (RHS) of this equation, $UB^{0.37}C^{0.62}$, the number of token-weighted updates (TWUs), given that it is the number of updates scaled by batch size and context size to incorporate the number of tokens seen at each update. Equation (6) suggests that the number of TWUs at which phase transition occurs can be expressed as a constant $T$ across model and training configurations. To further verify that this simple law indeed predicts the phase transition point of LMs trained with various training configurations, we can reformulate Equation (6) to predict the number of tokens $N$.

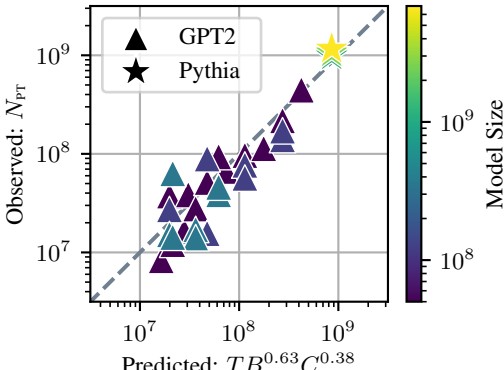

*Figure 3.* Predicted ($x$-axis) and observed ($y$-axis) numbers of pretraining tokens at which phase transition occurs. A strong correlation of $r = .986$ ($p < .001$) is found.

Given that $N = UBC$ by definition, we get:

$$N_{\text{PT}} = TB^{0.63}C^{0.38} \qquad (7)$$

The left hand side (LHS) of Equation (7) is the *observed* number of pretraining tokens at which phase transition occurs, and the RHS is the *predicted* point based on context size $C$ and batch size $B$, as well as the empirically found constant $T = e^{13.26}$. Now we can predict the number of tokens $N$ at which phase transition occurs, based on a constant $T$ and training configurations $C$ and $B$. In Figure 3, $x$-axis and $y$-axis correspond to the LHS and RHS, or the predicted and observed number of pretraining tokens at which phase transition occurs, respectively. We find a strong correlation of $r = .986$ ($p < .001$). This law holds for more than 3 orders of magnitude in model size (50M–7B), and 5 orders of magnitude in the number of tokens per update (500–2M).

### 6.2. Robustness

Given that the models presented thus far are based on a single random seed, and that the emergence point estimation through PWLF from each run could be noisy, we train with 2 additional random seeds the following 6 representative batch-context configurations: 16-64, 16-256, 16-1024, 16-2048, 128-1024, and 512-1024, which allows us to report the variability across 3 random seeds. We can see in Figure 4a that each configuration has tight 95% confidence intervals, suggesting robustness across different random seeds.

Similarly, we fit the predictive law, as described earlier (Section 6.1), on the set of 6 configurations for each of the 3 random seeds. Figure 4b shows that the predictive law $TB^{\beta+1}C^{\gamma+1}$ ($x$-axis) captures the observed emergence points ($y$-axis) well regardless of the random seed, showing its robustness. We report the full results of this uncertainty estimation in Appendix D.

**(a)** **(b)**

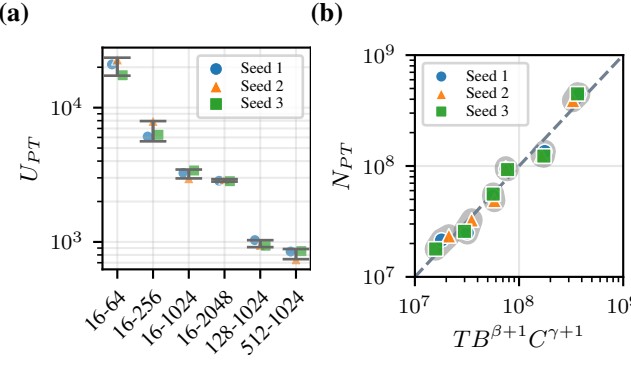

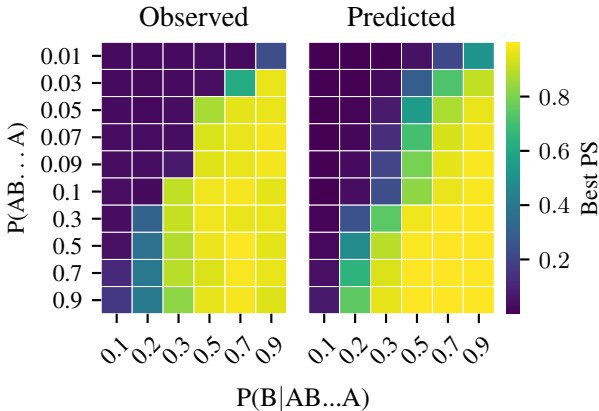

*Figure 4.* **(a)** IH emergence points measured in the number of updates and their 95% confidence intervals across 3 random seeds. Each configuration on the $x$-axis is a batch-context pair. **(b)** Predicted ($x$-axis) and observed ($y$-axis) numbers of pretraining tokens at which phase transition occurs. The color and shape of each point represent the random seed, and the gray shaded area represents data points from the same batch-context configuration across 3 different random seeds.

*Figure 5.* Best PS across all heads at the end of the training for each frequency reliability combination, observed (left) and predicted by the fitted model (right). Scores are represented in colors, with brighter colors representing higher scores.

# 7. Experiment 2: Semi-Natural Data Using Natural Bigram Statistics

## 7.1. Methods

In Figure 2 (right), we manipulated the repetition rate at the chunk level; in other words, we only manipulated the proportion of chunks with at least one bigram repetition. This repetitiveness at the chunk level is treated as a discrete variable called *burstiness* in Chan et al. (2022) and extended to a continuous variable in Reddy (2024) for an in-context classification task. Zucchet et al. (2025) also report the effect of repetition in training data; however, this is again at the sequence level and in an associative recall task, as opposed to language modeling task. To more precisely manipulate the repetition rate in the context of language modeling, we define two metrics, *frequency* and *reliability*.

*Frequency* measures the relative frequency at which $\langle A, B, \ldots, A \rangle$ is observed in a given dataset, expressed as $P(A, B, \ldots, A)$. It is simply the proportion of tokens in a given dataset that complete a $\langle A, B, \ldots, A \rangle$ sequence as the second occurrence of A, where $A \neq B$.

*Reliability* measures the conditional probability with which B is observed given $\langle A, B, \ldots, A \rangle$, expressed as $P(B \mid A, B, \ldots, A)$. It is the proportion of tokens in a given dataset that complete a $\langle A, B, \ldots, A, B \rangle$ sequence as the second occurrence of B, where $A \neq B$, divided by the aforementioned *frequency*. See Appendix F for more details on these two metrics, and Figure 10 for how these two metrics change for various context sizes in natural data.

To create training data that resemble natural data and also

satisfy desired values for these two metrics, we first generate a token-to-token transition matrix based on the bigram statistics from CC100. We then sample from this matrix, while imposing the specified frequency-reliability configuration, and train an LM for each configuration. See Appendix G for more details on the sampling procedure. We also note that the two properties are imposed on the second half of the sequence, rather than the entire sequence. We use the context size of 64 in this experiment.

## 7.2. Results

Each cell of Figure 5 represents an LM trained in the configuration specified by the $x$ and $y$ axes. We conduct a grid search over a search space defined by all possible combinations of each metric ranging from $\{0.1, 0.3, 0.5, 0.7, 0.9\}$. We initially found that the formation of IHs was insensitive to different values of $P(B \mid A, B, \ldots, A)$ when $P(A, B, \ldots, A) \geq 0.1$, and hence conducted an additional grid search over $P(A, B, \ldots, A) \in \{0.01, 0.03, 0.05, 0.07, 0.09\}$, as well as $P(B \mid A, B, \ldots, A) = 0.2$. With a total of 60 models colored based on the best PS in Figure 5, we can clearly see a decision boundary, where a decrease in either value will result in the failure of IH emergence.

Notably, LMs studied here seem to show stronger sensitivity to reliability than to frequency. In the bottom half of Figure 5, $[p_1, p_2]$ and $[p_2, p_1]$ do not always show the same result. For example, $P(A, B, \ldots, A) = 0.1$ always leads to IH formation except for $P(B \mid A, B, \ldots, A) \in \{0.1, 0.2\}$; however, it never forms when $P(B \mid A, B, \ldots, A) = 0.1$. This is notable, given that $[p_1, p_2]$ and $[p_2, p_1]$ have identical numbers of bigram repetitions because $P(A, B, \ldots, A, B) = P(B \mid A, B, \ldots, A)P(A, B, \ldots, A)$.

*Table 1.* Markov Processes used for pretraining data generation in Experiment 3. Each row represents a matrix that defines the Markov Process. The **Properties** column lists desired properties the matrix was optimized for, and the **Statistics** column summarizes the actual statistical properties each matrix had at the end of the optimization process. LD and CAT represents the binary variables local dependency and categoriality, respectively. H measures the entropy of the data, and KL divergence measures the fit between the desired distribution (as shown in the **Properties** column) and the actual marginal distribution of the generated matrix.

| | **Properties** | | | **Statistics** | | | |
| | Marginal | LD | CAT | $H(\cdot)$ | Intra-group | Inter-group | $D_{KL}(\cdot \parallel target)$ |
|---|---|---|---|---|---|---|---|
| $Zipf_{[+D+C]}$ | | ✓ | ✓ | 6.2142 | 0.3987 | 0.1007 | 0.0001 |
| $Zipf_{[+D-C]}$ | Zipfian | ✓ | ✗ | 6.1988 | 0.0999 | 0.1010 | 0.0001 |
| $Zipf_{[-D-C]}$ | | ✗ | ✗ | 9.5239 | 1 | 1 | - |
| $Unif_{[+D+C]}$ | | ✓ | ✓ | 6.5388 | 0.3719 | 0.0010 | 4e-6 |
| $Unif_{[+D-C]}$ | Uniform | ✓ | ✗ | 6.4759 | 0.0873 | 0.0104 | 0.0001 |
| $Unif_{[-D-C]}$ | | ✗ | ✗ | 13.2734 | 1 | 1 | - |
| $Gaus_{[+D+C]}$ | | ✓ | ✓ | 6.2435 | 0.3995 | 0.1003 | 0.0001 |
| $Gaus_{[+D-C]}$ | Gaussian | ✓ | ✗ | 6.2407 | 0.0999 | 0.1002 | 0.0001 |
| $Gaus_{[-D-C]}$ | | ✗ | ✗ | 12.3490 | 1 | 1 | - |

## 7.3. Predictability

To quantify the effect of frequency and reliability on the emergence of IHs, we fit a predictive model $\sigma(k(\alpha \log(P_A) + \beta \log(P_B) - \tau))$, where $P_A$ and $P_B$ are $P(A, B, \ldots, A)$ and $P(B \mid A, B, \ldots, A)$, respectively. We obtain $k = 4.299, \alpha = 0.472, \beta = 1.251, \tau = -2.322$, and the fitted model predicts the observed values well as shown in Figure 5 (right), with an MSE of 0.019. We can see that the best PS is more than twice as sensitive to reliability as it is to frequency, matching our earlier observation. Lastly, plugging in the fitted values and solving for $P_B$, we obtain a functional form: $P_B = 0.156 \times P_A{}^{-0.378}$.

## 8. Experiment 3: Synthetic Data

The previous experiment relied on a Markov Process obtained from a naturally occurring text (i.e., CC100). The main goal of this last experiment is to describe the properties of the underlying Markov Process necessary and/or sufficient for IH formation.

### 8.1. Methods

For simplicity, and to allow for more precise control over the data properties, we limit our scope to the second order Markov Process as the underlying generative process, which can be expressed as a token-to-token transition matrix $T \in \mathbb{R}^{|\mathcal{V}| \times |\mathcal{V}|}$. Once the desired properties (see below) are specified, we optimize the matrix using the Adam optimizer (see Appendix H for details). We consider three properties that we hypothesize to affect the formation of IHs: (1) local dependency, (2) categoriality, and (3) the shape of the marginal distribution. For (1) local dependency ($\pm$D), it is construed as: +D iff $P(w_{t+1}|w_t) \neq P(w_{t+1})$. In other words, unless the random variable $W$ is i.i.d. at each position $t$, we consider the distribution +D. This simply means that a distribution is +D if a word affects the next word.

For (2) categoriality ($\pm$C), as IHs have been shown to copy abstract patterns, such as semantic categories (e.g., color-object sequences; Olsson et al., 2022), we suspect that the presence of categories promotes the formation of IHs. To make this property compatible with the optimization process, we define categoriality by inter-group and within-group similarity scores. We define the presence (+C) and absence ($-$C) of categoriality as having within-category similarities of 0.4 and 0.1, respectively. Between-category similarity was always set to 0.1. We first assign $\frac{|\mathcal{V}|}{|\mathcal{C}|}$ tokens into each category $c \in \mathcal{C}$, thereby creating $|\mathcal{C}|$ groups with disjoint members. We then define inter-group similarity as the average cosine similarity between words (i.e., each of the $|\mathcal{V}|$ rows of the transition matrix $\mathcal{T} \in \mathbb{R}^{|\mathcal{V}| \times |\mathcal{V}|}$) from a given category $c$ and words from a different category $c'$:

$$\frac{1}{N} \sum_{c,c' \in \mathcal{C}, c \neq c'} \sum_{w \in c} \sum_{w' \in c'} sim(w, w') \qquad (8)$$

where $N$ is the number of such word pairs. Within-category similarity is likewise defined as the average similarity between all pairs of words that belong to the same category. If a distribution has a high within-category similarity but a lower across-category similarity, it means that categories exist in this distribution.

Lastly, for (3) marginal token distribution shape, we consider 3 distribution shapes that are increasingly less uniform: Uniform, Gaussian, and Zipfian distributions. This is because natural language is uniquely characterized by a Zipfian distribution, an inverse power law that expresses the frequency of a given word as inversely correlated with its

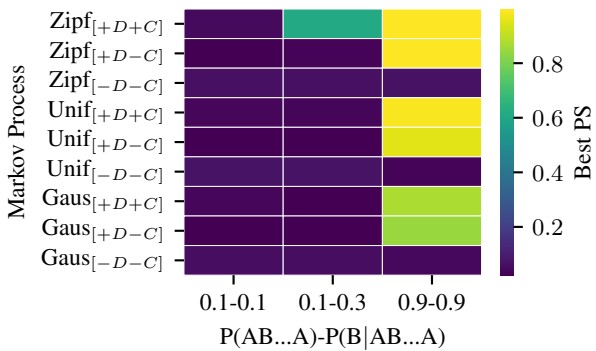

*Figure 6.* IH formation, as measured by PS, in each of the pretraining data generated by the Markov Processes. Each column represents a frequency-reliability configuration.

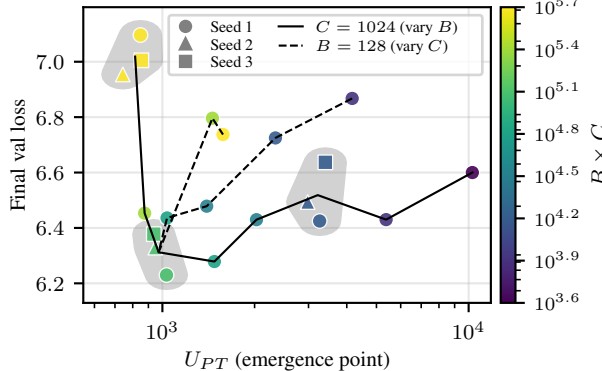

*Figure 7.* Final validation loss ($y$-axis) and the emergence point as measured in PS ($x$-axis). Colors represent the number of tokens seen per update. The solid line represents log-spaced batch sizes from 4 to 512 with the context size fixed at 1024, and the dashed line represents log-spaced context sizes from 64 to 4096 with the batch size fixed at 128. Same configurations run with three different seeds are grouped within gray shaded regions, and their centroids are used when connecting them to other points.

rank, and that it has been shown to affect the emergence of ICL capabilities (e.g., Chan et al., 2022).

Taken together, we have 3 (shape of marginal distribution) × 2 (local dependency) × 2 (categoriality) = 12 unique data configurations. We note that a subset of these configurations, specifically the −D+C configurations, are not conceivable. This is because, since −D is defined as a matrix with identical rows (each token's transition distribution is identical), both within-category and between-category similarities are 1. Hence, we have a total of 9 combinations of features, each of which generates the pretraining data.

Table 1 summarizes the properties as well as the actual statistics of each of these distributions. $H(\cdot)$ measures the conditional entropy of the distribution, estimated by taking the sum of the entropies of each row $\mathbf{r} \in \mathbb{R}^{|\mathcal{V}|}$, weighted by the stationary distribution (see Appendix I). Note that the last row of each block, marked by the −D configuration, has a higher entropy. This is because, for the −D configuration, each row of the matrix is identical to each other, and each word in this distribution is i.i.d. Hence, once the shape of the marginal distribution is specified, the entropy of this matrix is automatically determined. For example, Unif$_{[-D-C]}$ is by definition the maximally entropic distribution over $|\mathcal{V}|$ items: $\log_2 |\mathcal{V}|$. $D_{\text{KL}}(\cdot \parallel \text{target})$ is the KL divergence between the desired marginal distribution and the actual marginal distribution of the generated matrix. We can see that the divergence is very small for all distributions. Intra-group similarities, inter-group similarities, and $D_{\text{KL}}(\cdot \parallel \text{target})$ for the Dist$_{[-D-C]}$ configs are 1, 1, 0, respectively, by definition.

### 8.2. Results

In Figure 6, we first find that no property is by itself a *sufficient* condition for IH formation when measured by PS. For example, even under the highest bigram repetition condition (0.9-0.9), the −D configurations fail to produce IHs. Similarly, local dependency appears to be necessary in

this setup: no configurations with −D form IHs in Figure 6. Intuitively, if the next token distribution is independent of the current token, the model might not be learning to utilize past tokens in context to facilitate the next token prediction. We emphasize, however, that the result varies nontrivially when using different metrics to measure the emergence of IHs. As discussed in Section K.3, other metrics show partial IH-like behavior in some high-repetition −D configurations.

Second, as we showed in Section 7, we reconfirm that some level of bigram repetition is a *necessary* condition (Section 7). No configurations under the 0.1-0.1 column promote the formation of IHs. Lastly, interestingly, marginal distribution and categoriality seem to matter only when frequency and reliability are near the decision boundary. Whereas ±C and distribution shape do not affect IH formation for 0.1-0.1 and 0.9-0.9 columns, only Zipf$_{[+D+C]}$ results in the formation of IHs under the 0.1-0.3 column. The importance of skewed rank-frequency distributions, of which the Zipfian distribution is an example, is also reported in Chan et al. (2022); Reddy (2024).

## 9. Emergence and Training Dynamics

To better understand the role of emergence points in the context of LM pretraining dynamics in general, we plot the emergence points and final validation loss in Figure 7. Because models with a larger context size have an unfair advantage of larger test-time compute via attention, we feed the validation set in chunks of 64 tokens to all models. Hence, each model computes each token's loss using exactly the same amount of information regardless of the model configurations. From the solid line, which fixes the context size at 1024 and varies the batch size from 4 to 512 (dark

to bright), we can see that increasing the batch size moves the emergence point earlier as we have seen in Section 5, and also makes the validation loss lower until a certain point ($B = 64$) but higher beyond that point. We can also see that the emergence point does not change as much beyond $B = 128$. This is reminiscent of the critical batch size (CBS) hypothesis (McCandlish et al., 2018; Merrill et al., 2026): given a fixed compute, as we increase the batch size, the gain in the gradient quality diminishes, and the decrease in the number of updates results in a worse training outcome overall. In our case, we suspect that the emergence point will only become earlier with an increased batch size and better gradient update quality, and that once the gradient quality saturates, the emergence point will become constant. This V-shaped final validation loss curve is observed in the dashed line in Figure 7 as well, which varies context size while holding batch size fixed, suggesting the importance of context size in the CBS hypothesis.[4]

A related future direction is to distinguish between a mere timing shift and a qualitatively different downstream trajectory: that is, whether models in which IHs emerge earlier ultimately converge to similar solutions, or instead differ meaningfully in ICL behavior or other downstream measures. We leave a full characterization of such path dependence beyond the scope of the present study, which focuses on predicting when IHs emerge. Nevertheless, the non-monotonic relationship between emergence timing and final validation loss suggests that predicting IH emergence may be practically useful for choosing training configurations: earlier emergence is beneficial only up to a point, after which further accelerating emergence does not necessarily improve training outcomes.

## 10. Discussion

### 10.1. Importance of task selection

Our finding that larger context sizes promote earlier IH emergence may appear to contrast with Zucchet et al. (2025), who find later emergence with larger context sizes in an associative-recall task. We view these results as reflecting different effects of increasing context size. Larger contexts can simultaneously (i) increase the number of tokens per update, (ii) increase the number of repeated bigrams in natural language (Figure 9), and (iii) make retrieval harder by increasing query-key distance and the number of distractors. Our next-token-prediction setup appears to be dominated by the first two effects, while the retrieval-difficulty effect is more directly isolated in the associative recall setup in Zucchet et al. (2025); we discuss evidence for this effect in our setting in Appendix J. On the topic of task selection, it

is also worth noting that subword tokenization inflates the number of repeated bigrams in natural language, and using an orthographic tokenization might lead to different results, which is a promising direction for future research.

### 10.2. Implications for broader phase transition

Phase transition is a broad term that refers to any abrupt change in a target behavior or metric of interest. It has been studied in the context of $n$-gram distributions (Chang et al., 2024; Chang & Bergen, 2025; Michaelov et al., 2025), human language processing (Aoyama & Wilcox, 2025), and concept spaces in vision (Okawa et al., 2023; Park et al., 2024), to name a few. Previous studies find that IHs emerge after bigram learning (Olsson et al., 2022; Bietti et al., 2023). Although the present study focuses on IHs, and some aspects of the data properties may be specific to them (e.g., repetitions), a broader trend observed in Section 5 may generalize to other behavioral phases in LM pretraining. For example, Aoyama & Wilcox (2025) found a similar effect of batch size on the emergence of syntactic attention structure (Chen et al., 2024), and understanding to what extent the size-agnostic and TWU-dependent nature of IH emergence observed in this study also holds for other phenomena remains an important direction for future research.

### 10.3. Conclusion

We showed that the emergence of IHs can be predicted by batch size and context size (Section 6.1). We also showed that the frequency and reliability of bigram repetitions can express a decision boundary for IH formation (Section 7). Lastly, we found that, among local dependency, categoriality, the shape of the marginal distribution, frequency, and reliability, none of them alone was sufficient to ensure the formation of IHs when measured by PS (Section 8). We also find that local dependency coupled with high frequency and reliability always results in IH formation, and that the shape of the marginal distribution and categoriality appear to modulate IH formation when the frequency and reliability are near the decision boundary. We also showed that predicting IH emergence has practical implications for training efficiency given its non-monotonic relationship with the final validation loss under a fixed compute budget.

### 10.4. Limitations

As mentioned above, due to the limited compute resources, we only included models of up to 350M parameters in size for training, and 7B parameters for inference. Modern LMs are orders of magnitude larger in parameter count, and it is important to test if the trend holds for larger sizes. However, given a consistent trend we observed across three orders of magnitude (ranging from 50M GPT2 to 7B Pythia), we believe that a similar trend may hold for larger models.

---

[4]For an analysis on the potentially *later* emergence of IHs with a *larger* context size in Figure 7, see Appendix J.

## Impact Statement

This paper presents work whose goal is to advance the field of Machine Learning. There are many potential societal consequences of our work, none of which we feel must be specifically highlighted here.

## Acknowledgment

We thank the 4 anonymous reviewers, whose thoughtful and insightful comments substantially improved the paper. We also thank the organizers, reviewers, and participants of the CogInterp workshop at NeurIPS 2025, where we received invaluable feedback on the earlier version of this paper.

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

*Table 2.* List of hyperparameters used to train LMs.

| Architecture | | | |
|---|---|---|---|
| name | GPT2 50M | GPT2 125M | GPT2 350M |
| vocab_size | 50,257 | 50,257 | 50,257 |
| context_size | 4–2048 | 32–2048 | 32–2048 |
| d_embed | 768 | 1,024 | 1,024 |
| d_ffn | 3,072 | 4,096 | 4,096 |
| n_layer | 2 | 12 | 24 |
| n_head | 8 | 12 | 16 |
| activation | gelu | gelu | gelu |
| num_params | 50M | 125M | 350M |
| Training | | | |
| train_size | 1B | 1B | 1B |
| num_epoch | 1 | 1 | 1 |
| train_amount | 1B | 1B | 1B |
| batch_size | 4–512 | 16–256 | 16–256 |
| weight_decay | 0.1 | 0.1 | 0.1 |
| warmup_steps | 1% | 1% | 1% |
| lr | 5e-4 | 5e-4 | 5e-4 |
| lr_scheduler | cosine | cosine | cosine |

We also include 6 Pythia models: 70M, 160M, 410M, 1B, 2.8B, and 6.9B. For the details of these Pythia models, see Table 1 of Biderman et al. (2023).

## A. Determining the threshold for random attention

In Figure 2 (center), the lines representing context sizes of 4, 8, and 16 seem to be somewhat flat, meaning that the model does not improve in its ability to attend back to the token necessary to complete the repeated bigram. The high PSs associated with these models are simply due to the higher attention preceding tokens can get by chance; with the context size of 4, for example, the model has at most 4 tokens to attend back to, with the random attention of 0.25.

To systematically determine what counts as "above random," we simulate the random attention over previous tokens via Markov Chain Monte-Carlo (MCMC) using a Dirichlet distribution with a uniform prior. We find that attention weights as strong as 0.72, 0.35, and 0.16 are necessary for models with context sizes of 4, 8, and 16, respectively, to be considered *above random* at the alpha level of 0.01. Hence, we conclude that these three context sizes do not promote the formation of IHs, and in the remaining analyses in this subsection, we will focus on the rest of the models.

## B. Figure 2 plotted against the number of training tokens

In Figure 2, we plotted how batch size, context size, and bigram repetition rate affect the formation point of IHs, measured in the number of training steps. Trivially, models with different batch sizes and context sizes will have been exposed to different numbers of tokens at the same training step. This could raise the question of whether or not the observed shifting effect is just an artifact of the total number of training tokens being different on the same point on the $x$-axis. Hence, we show the same graph plotted against the number of total training tokens, instead of the number of training steps in Figure 8. Here again, we observe the shifting effect, but in the opposite direction: the smaller the batch/context size, the earlier the phase transition point. Therefore, we can say that, even when measured in the number of training tokens, IHs form at different points when we change batch size and/or context size, ruling out the possibility that the observed shifting effect in Figure 2 is an artifact of each point on the $x$-axis representing a different number of pretraining *tokens*.

## C. Fitting procedure for the prediction of emergence points

Here, we provide details on the procedure for fitting a regression model to predict emergence points.

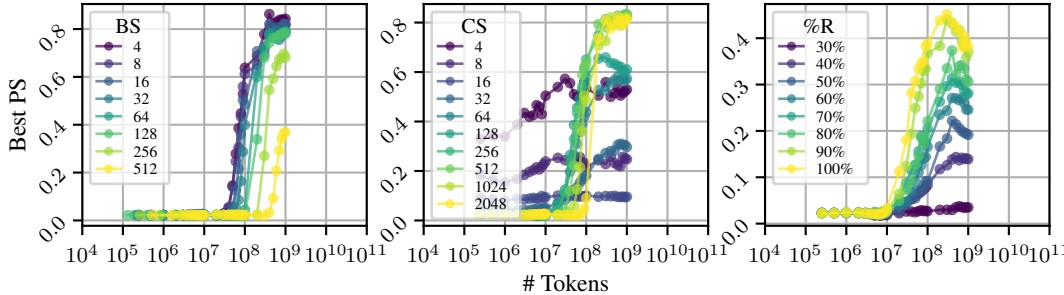

*Figure 8.* Developmental trajectories of PS of LMs with various batch sizes (left), context sizes (center), and repetitions (right) over the course of 1B tokens of pretraining, plotted against the number of updates (a) and the number of tokens (b). BS, CS, %R stand for batch size, context size, and the proportion of chunks with natural bigram repetitions, respectively.

### C.1. Emergence points

First, given a metric of interest that goes through an abrupt change, we need a way to systematically identify the point (in number of training steps or tokens) at which it is said to go through a phase transition point (or a certain capability measured by the score "emerges"). Prior work has used a few different approaches, such as the sharpest slope (Chen et al., 2024) and the first point at which the metric exceeds a given threshold (Aoyama & Wilcox, 2025). Here, given a curve that shows the development of a given metric's score over the course of some time metric, such as training steps and tokens (e.g., Figure 8), we fit a PWLF model to the curve with three segments. We choose three as we observe the initial stagnation phase, where the score is around 0, the abrupt improvement phase, and the eventual plateau phase. We then take the "knot" between the first two segments and operationalize that point as the emergence point.

### C.2. Data for fitting the linear regression models

The aforementioned emergence identification assigns a single emergence point for each model. Hence, we have as many data points as there are models. We have trained GPT2-50M models with 18 different batch and context size configurations (Section 5), as well as GPT2-125M and GPT2-350M with 7 different configurations each. However, because we have shown that emergence was not observed in the context sizes of {4, 8, 16}, we exclude those models in this analysis. We finally add 6 Pythia models {70M, 160M, 410M, 1B, 2.8B, 6.9B} all trained on batch size of 1024 and context size of 2048. In total, we fit a regression model on these 35 models. We reported the "fit" of this regression model trained on all of the data points; however, we also report the results from cross-fold validation as well in Table 5.

### C.3. Regression model details for phase transition prediction

In Section 5, we fitted a linear regression in log space using batch size, context size, and model size as predictors for the phase transition point.

Table 3 summarizes the linear regression model with all three predictors. A few consistent trends emerge: first, batch size and context size are (1) statistically significant across the board, regardless of the metric used to define the phase transition point, and (2) in the negative direction, meaning that the increase in these predictors leads to *earlier* emergence. Second, model size is non-significant across the board, regardless of the metric. Hence, we conclude that the emergence points are independent of model size. The final regression models without the model size are summarized in Table 4.

Lastly, as mentioned earlier, we also report the results of 5-fold cross-validation of the full regression models and the models without model size in Table 5. Each fold trains the regression model on 80% of the training data (28 models) and predicts the emergence points of the remaining 20% of the data (7 models). Most test $R^2$s range from 0.8 to 0.9, indicating an excellent fit to the *held-out* data points. Note that $R^2_{test}$ is computed as:

$$1 - \frac{SS_{residual}}{SS_{total}} \tag{9}$$

and the value of 1 indicates a perfect prediction on the test set.

*Table 3.* A summary of linear regression analyses with an intercept, logB (batch size), logC (context size), and logM (model size) as independent variables and the phase transition point (in the number of updates) in each of the four metrics, PS, LA, AR (accuracy), AR (mean rank), as dependent variables. Significantly **positive** and **negative** test statistics are colored in **green** and **red**, respectively.

| y | x | coef | std err | t | p | 95% CI | |
|---|---|---|---|---|---|---|---|
| **PS** | **Intercept** | **13.2700** | 0.468 | **28.326** | 0.000 | **12.315** | **14.225** |
| | **log(B)** | **-0.3683** | 0.044 | **-8.318** | 0.000 | **-0.459** | **-0.278** |
| | **log(C)** | **-0.6149** | 0.073 | **-8.411** | 0.000 | **-0.764** | **-0.466** |
| | **log(M)** | -0.0027 | 0.063 | -0.043 | 0.966 | -0.130 | 0.125 |
| **LA** | **Intercept** | **13.6083** | 0.477 | **28.529** | 0.000 | **12.635** | **14.581** |
| | **log(B)** | **-0.3735** | 0.045 | **-8.286** | 0.000 | **-0.465** | **-0.282** |
| | **log(C)** | **-0.6461** | 0.074 | **-8.678** | 0.000 | **-0.798** | **-0.494** |
| | **log(M)** | -0.0069 | 0.064 | -0.108 | 0.915 | -0.137 | 0.123 |
| **AR (accuracy)** | **Intercept** | **16.3236** | 0.632 | **25.846** | 0.000 | **15.036** | **17.612** |
| | **log(B)** | **-0.5885** | 0.060 | **-9.859** | 0.000 | **-0.710** | **-0.467** |
| | **log(C)** | **-0.7925** | 0.099 | **-8.040** | 0.000 | **-0.993** | **-0.591** |
| | **log(M)** | -0.0154 | 0.084 | -0.182 | 0.856 | -0.188 | 0.157 |
| **AR (mean rank)** | **Intercept** | **12.3780** | 0.723 | **17.125** | 0.000 | **10.904** | **13.852** |
| | **log(B)** | **-0.5194** | 0.068 | **-7.603** | 0.000 | **-0.659** | **-0.380** |
| | **log(C)** | **-0.4928** | 0.113 | **-4.369** | 0.000 | **-0.723** | **-0.263** |
| | **log(M)** | 0.1295 | 0.097 | 1.340 | 0.190 | -0.068 | 0.327 |

*Table 4.* A summary of linear regression analyses with an intercept, logB (batch size), and logC (context size) as independent variables and the phase transition point (in the number of updates) in each of the four metrics, PS, LA, AR (accuracy), AR (mean rank), as dependent variables. Significantly **positive** and **negative** test statistics are colored in **green** and **red**, respectively.

| y | x | coef | std err | t | p | 95% CI | |
|---|---|---|---|---|---|---|---|
| **PS** | **Intercept** | **13.2630** | 0.432 | **30.666** | 0.000 | **12.382** | **14.144** |
| | **logB** | **-0.3689** | 0.041 | **-9.010** | 0.000 | **-0.452** | **-0.286** |
| | **logC** | **-0.6155** | 0.071 | **-8.694** | 0.000 | **-0.760** | **-0.471** |
| **LA** | **Intercept** | **13.5904** | 0.440 | **30.856** | 0.000 | **12.693** | **14.488** |
| | **logB** | **-0.3752** | 0.042 | **-8.999** | 0.000 | **-0.460** | **-0.290** |
| | **logC** | **-0.6475** | 0.072 | **-8.981** | 0.000 | **-0.794** | **-0.501** |
| **AR (accuracy)** | **Intercept** | **16.2837** | 0.583 | **27.913** | 0.000 | **15.095** | **17.472** |
| | **logB** | **-0.5922** | 0.055 | **-10.723** | 0.000 | **-0.705** | **-0.480** |
| | **logC** | **-0.7957** | 0.095 | **-8.333** | 0.000 | **-0.990** | **-0.601** |
| **AR (mean rank)** | **Intercept** | **12.7138** | 0.686 | **18.525** | 0.000 | **11.316** | **14.112** |
| | **logB** | **-0.4880** | 0.065 | **-7.511** | 0.000 | **-0.620** | **-0.356** |
| | **logC** | **-0.4656** | 0.112 | **-4.145** | 0.000 | **-0.694** | **-0.237** |

*Table 5.* Cross-fold validation results for the full regression model (left) and the regression model without model size (right). Each fold trains the regression model on 80% of the data and predicts on the remaining 20% of the data. Recall that $\beta, \gamma, \theta$ are coefficients of batch size, context size, and model size, respectively. Prediction quality is measured in $R^2_{test} = \frac{SS_{residual}}{SS_{total}}$

| Metric | Fold | Full | | | | w/o Model Size | | |
|---|---|---|---|---|---|---|---|---|
| | | $\beta$ | $\gamma$ | $\theta$ | $R^2_{test}$ | $\beta$ | $\gamma$ | $R^2_{test}$ |
| **PS** | 1 | -0.3663 | -0.6223 | 0.0130 | 0.7629 | -0.3617 | -0.6204 | 0.7653 |
| | 2 | -0.3517 | -0.5380 | -0.0055 | 0.8969 | -0.3527 | -0.5387 | 0.8966 |
| | 3 | -0.3607 | -0.6232 | 0.0033 | 0.9282 | -0.3600 | -0.6222 | 0.9284 |
| | 4 | -0.4000 | -0.6023 | 0.0026 | 0.8338 | -0.3992 | -0.6019 | 0.8340 |
| | 5 | -0.3775 | -0.6438 | -0.0167 | 0.8204 | -0.3811 | -0.6478 | 0.8234 |
| **LA** | 1 | -0.3713 | -0.6599 | 0.0101 | 0.7629 | -0.3677 | -0.6584 | 0.7649 |
| | 2 | -0.3627 | -0.5386 | -0.0127 | 0.8871 | -0.3649 | -0.5403 | 0.8864 |
| | 3 | -0.3724 | -0.6585 | -0.0004 | 0.9056 | -0.3725 | -0.6586 | 0.9055 |
| | 4 | -0.3933 | -0.6364 | -0.0032 | 0.8818 | -0.3943 | -0.6369 | 0.8816 |
| | 5 | -0.3829 | -0.6811 | -0.0171 | 0.8073 | -0.3866 | -0.6852 | 0.8092 |
| **AR (accuracy)** | 1 | -0.5963 | -0.7979 | 0.0176 | 0.7907 | -0.5900 | -0.7952 | 0.7942 |
| | 2 | -0.5866 | -0.6605 | -0.0283 | 0.9060 | -0.5916 | -0.6643 | 0.9055 |
| | 3 | -0.5762 | -0.8585 | 0.0488 | 0.8115 | -0.5670 | -0.8437 | 0.8257 |
| | 4 | -0.6115 | -0.7897 | -0.0189 | 0.9417 | -0.6171 | -0.7924 | 0.9418 |
| | 5 | -0.5897 | -0.7847 | -0.1022 | 0.8923 | -0.6117 | -0.8093 | 0.9353 |
| **AR (mean rank)** | 1 | -0.5660 | -0.4922 | 0.1891 | 0.7325 | -0.4989 | -0.4634 | 0.8427 |
| | 2 | -0.4967 | -0.5753 | 0.1073 | 0.7413 | -0.4778 | -0.5608 | 0.7146 |
| | 3 | -0.5175 | -0.4716 | 0.1340 | 0.8988 | -0.4922 | -0.4309 | 0.8876 |
| | 4 | -0.5770 | -0.4459 | 0.1659 | 0.8104 | -0.5273 | -0.4216 | 0.8597 |
| | 5 | -0.4477 | -0.4918 | 0.0377 | 0.6032 | -0.4396 | -0.4827 | 0.5866 |

## D. Variability across random seeds

We train a subset of 6 batch-context configurations with 2 additional random seeds to quantify the variability across 3 random seeds. Table 6 (left) summarizes the mean IH emergence points, as measured in the number of updates $U_{PT}$, and their standard deviations. Table 6 (right) summarizes the mean fitted parameter values for the predictive law described in Equation (5): $U_{\text{PT}} = e^{\alpha} B^{\beta} C^{\gamma}$.

*Table 6.* Uncertainty estimation of emergence points (left) and predictive law parameters (right). Configurations are presented in (batch, context) notation.

| | IH emergence point $\log_{10}(U_{PT})$ | | | | | | Predictive law parameters | | | |
|---|---|---|---|---|---|---|---|---|---|---|
| | (16,64) | (16,256) | (16,1024) | (16,2048) | (128,1024) | (512,1024) | $\alpha$ | $\beta$ | $\gamma$ | $R^2$ |
| Mean | 4.308 | 3.828 | 3.507 | 3.458 | 2.987 | 2.910 | 13.540 | -0.470 | -0.589 | 0.968 |
| SD | 0.060 | 0.064 | 0.029 | 0.008 | 0.023 | 0.034 | 0.401 | 0.026 | 0.048 | 0.005 |

## E. Distribution of bigram repetitions with various context sizes

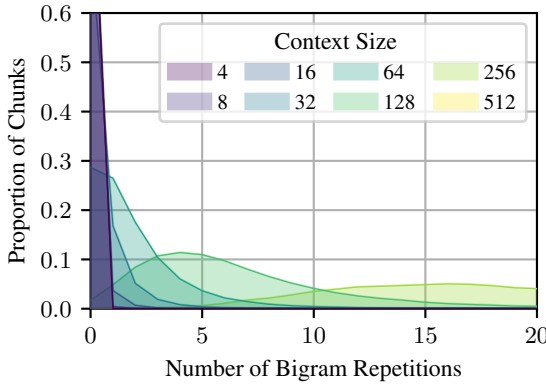

*Figure 9.* Smoothed distribution of chunks with various numbers of bigram repetitions. Context sizes 1024 and 2048 are not visible, and hence removed from the plot. The plot is truncated at y=0.6 for readability, but context sizes of 4, 8, and 16 had >95% of chunks with no bigram repetitions.

Since adding or removing bigrams in naturally occurring texts introduces noise, such as broken syntax, we manipulate the frequency of repeated bigrams in natural language data by first putting tokenized texts into chunks of $c$, where $c$ is the context size of the LM, and then selecting those natural chunks to ensure $p\%$ of the chunks of the resulting training data include no bigram repetition at all, where $p$ is the parameter we can control. Modern LMs have a context size of at least 1024. However, in naturally occurring texts, sequences of 1024 tokens without any repeated bigrams $\langle A, B, \ldots, A, B \rangle$ in them are very rare, if not non-existent. In general, as shown in Figure 9, larger context size trivially tends to contain more repeated bigrams, and we find that the context size of 64 strikes a balance between including enough context and containing a good number of chunks with and without repeated bigrams. Hence, in Section 7 and Section 8, we use LMs with a context size of 64.

## F. Frequency $P(A, B, \ldots, A)$ and Reliability $P(B \mid A, B, \ldots, A)$

Recall that Elhage et al. (2021); Olsson et al. (2022) define IHs as specific heads that complete the repeated $\langle A, B, \ldots, A, B \rangle$ sequence when seeing $\langle A, B, \ldots, A \rangle$. As a naive hypothesis, we speculate that the presence of repeated bigrams (separated by an arbitrarily long sequence of tokens within the LM's context size) is essential for IH formation. Consider the following sequence, which we will use as a running example: In this example, there are 4 tokens that occur more than once (i.e., A, B, E, F). For the first occurrence of these 4 tokens, each of them is followed by *something*; for example, A is followed by B, and B is followed by C. For the second occurrence of these 4 tokens, one can induce that a bigram tends to repeat *in-context*, if these 4 tokens are reliably followed by what followed them the first time they occurred; namely, if A, B, E,

*Table 7.* $R_U$ and $R_B$ of an example sequence. This sequence has a $P(R_U) = \frac{4}{10}$ and $P(R_B) = \frac{2}{10}$, hence **frequency** and **reliability** are 0.4 and 0.5, respectively.

| Word | A | B | C | D | E | F | A | B | E | F |
|---|---|---|---|---|---|---|---|---|---|---|
| $R_U$ | 0 | 0 | 0 | 0 | 0 | 0 | 1 | 1 | 1 | 1 |
| $R_B$ | 0 | 0 | 0 | 0 | 0 | 0 | 1 | 0 | 1 | 0 |

F, are followed by B, C, F, A, respectively. In Table 7, there exist 4 such opportunities for learning the bigram repetition, and 2 such opportunities actually reward such learning, since only the second occurrences of A and E are followed by the same bigram continuations of their first occurrences. Here, we have touched upon the two knobs we aim to define here. **Frequency** involves the tokens that are of the word type occurring for the $n$-th time where $n \geq 2$ (A, B, E, F in Table 7). **Reliability** measures, of all such tokens, how many of them actually complete the same bigram continuation (B, E in Table 7). We will now define each of these two measures formally.

**Frequency**: given a binary variable *unigram repetition* $R_U \in \{0, 1\}$, whose value is 1 if a given token is the second occurrence of A of the $\langle A, B, \ldots, A \rangle$ sequence (i.e., a repeated unigram), and 0 otherwise, we define "frequency" as $P(R_U)$. Informally, in induction terms, this is $P(A, B, \ldots, A)$, or the rate at which $\langle A, B, \ldots, A \rangle$ sequence is encountered. In practice, this is equivalent to the sum of all unigram counts in a given sequence, with the first occurrence removed:

$$P(A, B, \ldots, A) = \frac{1}{|\mathbf{s}|} \sum_{w \in \mathcal{V}} \max(c(w, \mathbf{s}) - 1, 0) \tag{10}$$

where $\mathcal{V}$ is the vocabulary, or the set of all possible unigrams, and $c(w, \mathbf{s})$ is a count of a word $w$ in a sequence $\mathbf{s}$ of the same size as the model's context size. In Table 7, we saw 4 such tokens (tokens where $R_U = 1$: A, B, E, F) in the context of size 10, hence $P(R_U) = P(A, B, \ldots, A) = \frac{4}{10}$. Technically, consecutive occurrences of a given token type should not be counted; however, this does not happen frequently in the real data, and no tokens were allowed to occur twice in a row in synthetic data, trivializing this problem (see Algorithm 1).

**Reliability**: given a binary variable *bigram repetition* $R_B \in \{0, 1\}$, whose value is 1 if a given token's *next token* is the second occurrence of B of the $\langle A, B, \ldots, A, B \rangle$ sequence (i.e., a repeated bigram), and 0 otherwise, we define "reliability" as $\frac{P(R_B)}{P(R_U)}$. Informally, in induction term, $P(R_B)$ is $P(A, B, \ldots, A, B)$, and reliability, or $\frac{P(R_B)}{P(R_U)}$, is equivalent via the chain rule to the conditional probability $P(B \mid A, B, \ldots, A)$: the rate at which an $\langle A, B, \ldots, A \rangle$ sequence is followed by B:

$$P(B \mid A, B, \ldots, A) = \frac{P(A, B, \ldots, A, B)}{P(A, B, \ldots, A)} = \frac{P(R_B)}{P(R_U)} = \frac{\frac{1}{|\mathbf{s}|} \sum_{b \in \mathcal{B}} \max(c(b, \mathbf{s}) - 1, 0)}{\frac{1}{|\mathbf{s}|} \sum_{w \in \mathcal{V}} \max(c(w, \mathbf{s}) - 1, 0)} \tag{11}$$

where $\mathcal{B}$ denotes the set of all possible bigrams, and $c(b, \mathbf{s})$ the count of the bigram $b$ in a given sequence $\mathbf{s}$. In Table 7, of all the 10 positions (tokens), 2 completed the $\langle A, B, \ldots, A, B \rangle$ sequence (tokens where $R_B = 1$: second occurrences of B and F), hence $P(R_B) = P(A, B, \ldots, A, B) = \frac{2}{10}$. Therefore, reliability is $\frac{2}{10} \div \frac{4}{10} = \frac{1}{2}$. It might make more intuitive sense to compute this directly without using the chain rule: of all the 4 tokens that complete the $\langle A, B, \ldots, A \rangle$ sequence, 2 of them are followed by B, hence $\frac{2}{4} = \frac{1}{2}$. Equivalently, of the tokens where $R_U = 1$ in Table 7, half of them also have $R_B = 1$. However, for computational purposes, the chain rule is much simpler, which is the reason we introduced the chain rule-based calculation above. In this study, "frequency" and $P(A, B, \ldots, A)$ are used interchangeably, and so are "reliability" and $P(B \mid A, B, \ldots, A)$.

Note that this is a more precise characterization of what Chan et al. (2022) called "burstiness." In their formulation, where the task was to predict the label of an image given image-label pairs in-context, a "bursty" sequence contains certain image-label pairs more often than others, while controlling for the marginal distribution over all sequences. This is effectively equivalent to increasing both frequency and reliability in our terms. Two sequences of image-label pairs $(I_i, L_i)$: $\langle I_1, L_1, I_2, L_2, I_3, L_3, I_4, L_4 \rangle$ (non-bursty) and $\langle I_1, L_1, I_2, L_2, I_1, L_1, I_3, L_3 \rangle$ (bursty) can be reformulated as $\langle A, B, C, D, E, F, G, H \rangle$ and $\langle A, B, C, D, A, B, E, F \rangle$, respectively, and the former has frequency and reliability of 0, whereas the latter has frequency and reliability of 1/8 and 1/2 ($\langle A, B, \ldots, A \rangle$ is followed by B, but $\langle B, C, \ldots, B \rangle$ is not followed by C), respectively. We can verify that natural texts with various chunk sizes are not only different in terms of the number of repeated bigrams (as shown in Figure 9), but also in the two probabilities defined above. Figure 10 confirms this: both $P(A, B, \ldots, A)$ and $P(B \mid A, B, \ldots, A)$ increase as the context size increases. While $P(A, B, \ldots, A)$ seems to

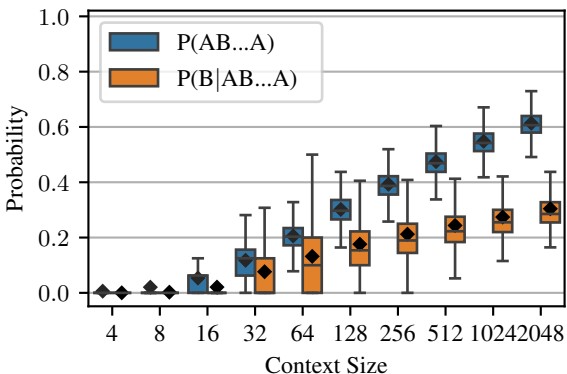

*Figure 10.* Distribution of chunks with various $P(A, B, \ldots, A)$ and $P(B \mid A, B, \ldots, A)$ for each context size. Only the quartile box, median (center line in each box), mean (diamond), and whiskers are shown, and outliers are not shown for readability.

increase log-linearly ($x$-axis is on a log scale), the increase in $P(B \mid A, B, \ldots, A)$ seems to slow down. Since IHs were not forming, or very weak at most, for a context size of 32, we speculate that the threshold values of $P(A, B, \ldots, A)$ and $P(B \mid A, B, \ldots, A)$ are somewhere between 0.1–0.2, and 0.1–0.15, respectively.

## G. Algorithm for Frequency- and Reliability-Constrained Training Data Generation

To fully control the two properties, frequency $P(A, B, \ldots, A)$ and reliability $P(B \mid A, B, \ldots, A)$ in text data, we need a way to sample words from some distribution, while enforcing desired values of these two knobs. To this end, we approximate natural language by first tokenizing texts from the English subcorpus of the Common Crawl Corpus (CC100; Conneau et al., 2020; Wenzek et al., 2020) and collecting token bigram statistics. We then create a token-to-token transition matrix $T \in \mathbb{R}^{|\mathcal{V}| \times |\mathcal{V}|}$, where $\mathcal{V}$ is the vocabulary. We use an off-the-shelf GPT2 tokenizer, but reduce the vocabulary size to 10,000.

In Algorithm 1, we outline the semi-synthetic data generation algorithm, where $(\mathcal{A} \backslash \mathcal{B})$ denotes an asymmetric difference, or a set of elements in $\mathcal{A}$ but not in $\mathcal{B}$, $\{x \mid x \in \mathcal{A} \text{ and } x \notin \mathcal{B}\}$. $\mathcal{U}_{<t-1}$ is a set of unigrams attested before time step $t-1$, and $\mathcal{B}_x$ denotes a set of attested bigram continuations of token $x$. The idea is that we first generate the first half of the sequence by randomly walking through the Markov chain. This is to ensure that a sufficient number of unique tokens are present in the sequence before the restricted generation can take place; otherwise, Algorithm 1 often produces degenerate sequences especially with high values of $P(A, B, \ldots, A)$ and $P(B \mid A, B, \ldots, A)$. For the second half, based on the condition (if the prefix constitutes $\langle A, B, \ldots, A \rangle$) and constraints (to make a new $\langle A, B, \ldots, A \rangle$ and/or $\langle A, B, \ldots, A, B \rangle$), the word at each time step $w_t$ is sampled from the distribution $\mathcal{D}$ restricted to some subset $\mathcal{S}$ that satisfies the conditions and constraints. For simplicity, no tokens were allowed to occur consecutively.

Note that the constraints $P(A, B, \ldots, A)$ and $P(B \mid A, B, \ldots, A)$ are imposed at each token generation step uniformly, meaning that we do not devise a token-specific repetition rate. It is an open question whether or not this is a sensible simulation of the token distribution in natural language: at its core, it comes down to the question of to what extent each word in natural language differs in its probability of participating in repeated bigrams beyond their differences in unigram and bigram distributions, and we will leave this to future work.

## H. Matrix Optimization

In Section 8, we defined the underlying Markov Processes as transition matrices, which were then used to generate synthetic data, via a set of desired properties. These transition matrices were optimized using the Adam optimizer to satisfy the desired properties as closely as possible.

$$\mathcal{L} = \lambda_1 \mathcal{L}_D + \lambda_2 \mathcal{L}_E + \lambda_3 \mathcal{L}_P + \lambda_4 \mathcal{L}_{WC} + \lambda_5 \mathcal{L}_{WA} \tag{12}$$

where the 5 terms correspond to distribution loss, entropy loss, peakedness loss, within-category loss, and across-category loss, respectively. The 5 $\lambda$s correspond to the weighting factors.

**Distribution loss ($\mathcal{L}_D$).** This term is to ensure the transition matrix $T \in \mathbb{R}^{|\mathcal{V}| \times |\mathcal{V}|}$ has a marginal distribution of the desired

---

**Algorithm 1** Corpus generation constrained on $P(\mathsf{A}, \mathsf{B}, \ldots, \mathsf{A})$ and $P(\mathsf{B} \mid \mathsf{A}, \mathsf{B}, \ldots, \mathsf{A})$

---

    *Sample a token sequence* $\mathbf{s}$ *from the distribution* $\mathcal{D} : \{w \mapsto \mathcal{D}_w \in \mathbb{R}^{|\mathcal{V}|} \mid w \in \mathcal{V}\}$,
    *constrained on* $\alpha = P(\mathsf{A}, \mathsf{B}, \ldots, \mathsf{A})$ *and* $\beta = P(\mathsf{B} \mid \mathsf{A}, \mathsf{B}, \ldots, \mathsf{A})$

1: **Input:** context_size, $\mathcal{V}, \mathcal{D} \in \mathbb{R}^{|\mathcal{V}| \times |\mathcal{V}|}, \alpha, \beta$
2: $\mathbf{s} \leftarrow []$                                                      ▷ init a token sequence
3: $\mathcal{U}_t \leftarrow \{\}$                                                 ▷ init a unigram set at step $t$
4: $\mathcal{B}_t \leftarrow \{\}$                        ▷ init a bigram continuation dict $\{w \mapsto \mathcal{B}_t(w) \mid w \in \mathcal{U}_t\}$ at step $t$
5: $m \leftarrow$ context_size$//2$
6: **for** $t$ from 1 to $m$ **do**                                             ▷ first half
7:     $w_t \sim \mathcal{D}_{w_{t-1}}(\mathcal{V} \backslash \mathcal{U}_{<t})$
8:     add $w_t$ to $\mathbf{s}$
9:     update $\mathcal{U}_t$ and $\mathcal{B}_t$
10: **for** $t$ from $m$ to context_size **do**                                  ▷ second half
11:     is_aba $\leftarrow w_t$ in $\mathcal{U}_{t-1}$
12:     make_aba $\leftarrow$ random.random() $\leq \alpha$                        ▷ $\alpha$ represents $P(\mathsf{A}, \mathsf{B}, \ldots, \mathsf{A})$
13:     make_abab $\leftarrow$ random.random() $\leq \beta$                   ▷ $\beta$ represents $P(\mathsf{B} \mid \mathsf{A}, \mathsf{B}, \ldots, \mathsf{A})$
14:     **if** is_aba **then**
15:       **if** make_abab **then**
16:         $w_t \sim \mathcal{D}_{w_{t-1}}(\cdot \mid \mathcal{B}_t(w_{t-1}))$
17:       **else**
18:         **if** make_aba **then**
19:           $w_t \sim \mathcal{D}_{w_{t-1}}(\cdot \mid \mathcal{U}_{<t} \backslash \mathcal{B}_t(w_{t-1}))$
20:         **else**
21:           $w_t \sim \mathcal{D}(\cdot \mid \mathcal{V} \backslash \{\mathcal{U} \cup \mathcal{B}_t(w_{t-1})\})$
22:     **else**
23:       **if** make_aba **then**
24:         $w_t \sim \mathcal{D}_{w_{t-1}}(\cdot \mid \mathcal{U}_{<t})$
25:       **else**
26:         $w_t \sim \mathcal{D}_{w_{t-1}}(\cdot \mid \mathcal{V} \backslash \mathcal{U}_{<t})$
27:     add $w_t$ to $\mathbf{s}$
28:     update $\mathcal{U}_t$ and $\mathcal{B}_t$

---

shape (Uniform, Gaussian, or Zipfian, as discussed in Section 8). We penalize the divergence from the desired shape by including KL divergence between the actual marginal distribution and the desired distribution:

$$\mathcal{L}_D = \sum_{i=1}^{|\mathcal{V}|} P(w_i) \log \frac{P(w_i)}{Q(w_i)}, \tag{13}$$

where P and Q are desired and actual marginal distributions, respectively.

**Entropy loss ($\mathcal{L}_E$).** This term is to ensure each transition matrix is comparable in predictability. Because a transition matrix sampled from natural language had the entropy of $\approx 6.2$, we set the target entropy value to be 6.2, and included the squared difference as a loss term:

$$\mathcal{L}_E = \|H_{\text{target}} - H(T)\|_2^2, \tag{14}$$

where the estimation of $H(T)$ is detailed in Appendix I.

**Peakedness loss ($\mathcal{L}_P$).** We find that the matrix optimization often suffers from a degenerate matrix, where the desired properties are satisfied by allocating a very large probability mass to a single token in each row. To mitigate this problem, we include the mean of row-wise maximum probabilities:

$$\mathcal{L}_P = \frac{1}{|\mathcal{V}|} \sum_{i=1}^{|\mathcal{V}|} \max T_{i,:}, \tag{15}$$

where $T_{i,:}$ is the $i$-th row of the matrix $T$.

**Within-category loss ($\mathcal{L}_{WC}$) and across-category loss ($\mathcal{L}_{AC}$).** As defined in Section 8, the within-category similarity is the mean similarity of all pairs of words within a category, whereas the across-category similarity is the mean similarity of all pairs of words from different categories. For the +C configuration, we set the target within- and across-category similarities to be 0.4 and 0.1, respectively, and both were set to be 0.1 for the −C configuration. The squared differences between the actual and desired within-/across-category similarities were included as loss terms. The within-category loss is defined as:

$$\mathcal{L}_{WC} = \|\text{WC}_{\text{target}} - \text{WC}(T)\|_2^2, \tag{16}$$

where WC is within-category similarity:

$$\frac{1}{N} \sum_{c \in C} \sum_{w,w' \in c, w \neq w'} sim(w, w') \tag{17}$$

Similarly, across-category loss is defined as:

$$\mathcal{L}_{AC} = \|\text{AC}_{\text{target}} - \text{AC}(T)\|_2^2, \tag{18}$$

where AC is across-category similarity:

$$\frac{1}{N} \sum_{c,c' \in C, c \neq c'} \sum_{w \in c} \sum_{w' \in c'} sim(w, w') \tag{19}$$

We optimize the matrix with these loss terms for 5,000 steps with $\lambda_1 = 100$, $\lambda_2 = 0.01$, $\lambda_3 = 0.1$, $\lambda_4 = \lambda_5 = 5$.

## I. Conditional Entropy

Conditional entropy is defined as:

$$H(X \mid Y) = \sum_{y \in Y} P(y) \left[ \sum_{x \in X} P(x \mid y) \log \frac{1}{P(x \mid y)} \right] \tag{20}$$

For a transition matrix $T \in \mathbb{R}^{|\mathcal{V}| \times |\mathcal{V}|}$, where $\mathcal{V}$ is the vocabulary, it can be expressed as:

$$H(T) = \sum_{i=1}^{|\mathcal{V}|} P(w_i) \left[ \sum_{j=1}^{|\mathcal{V}|} P(w_j \mid w_i) \log \frac{1}{P(w_j \mid w_i)} \right] \tag{21}$$

Since the transition matrix $T$ is a row-stochastic matrix, and each row sums to 1, the conditional probability $P(w_j|w_i)$ is an entry $T[i, j]$. The marginal probability $P(w_i)$ can be estimated by obtaining the stationary distribution of the transition distribution $T$, which is a left eigenvector with the eigenvalue of 1. Assume a ground-truth stationary distribution $\pi$; this stationary distribution, which is the unigram distribution the transition matrix converges to, should remain unchanged after transitions:

$$\pi T = \pi \tag{22}$$

An eigenvector is a vector that only gets scaled by a factor $\lambda$ after a linear transformation $L$. Hence, we can find the stationary unigram distribution $\pi$ by finding the left eigenvector with the eigenvalue of 1 of a linear transformation $T$.

Now that we have the stationary distribution $\pi$, Equation (21) can be expressed as a matrix multiplication:

$$H(T) = \sum_{i=1}^{|\mathcal{V}|} \pi[i] \cdot H(T[i,]) \tag{23}$$

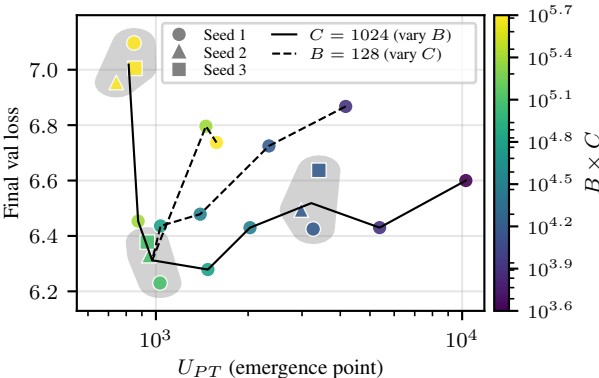

*Figure 11.* Figure 7 reproduced for readability. Final validation loss ($y$-axis) and the emergence point as measured in PS ($x$-axis). Colors represent the number of tokens seen per update. The solid line represents log-spaced batch sizes from 4 to 512 with the context size fixed at 1024, and the dashed line represents log-spaced context sizes from 64 to 4096 with the batch size fixed at 128. Same configurations run with three different seeds are grouped within gray shaded regions, and their centroids are used when connecting them to other points.

## J. Context size and retrieval difficulty

In Section 9, we mainly focus on the observation that both batch size and context size participate in the V-shaped validation loss curve (Figure 11). However, the picture with the context size seems more complicated. Unlike batch size, where an increase always results in an earlier emergence point, increasing context size could potentially make the emergence later. This is due to an interplay of (at least) 3 effects of increasing the context size: (1) more tokens seen per update ("shifting" in Section 5), (2) more repetitions per sequence ("slanting" in Section 5), and (3) higher retrieval difficulty. The CBS hypothesis suggests that the improvement in gradient quality in (1) saturates, and we suspect that the increase in frequency and reliability in (2) also does (e.g. increase in $P(\mathsf{B} \mid \mathsf{A}, \mathsf{B}, \ldots, \mathsf{A})$ slows down with larger context sizes in Figure 9). Zucchet et al. (2025) show that the third effect, the retrieval difficulty, increases with a larger context size, which delays the emergence. We surmise that, with the saturation of (1) and (2), the effect (3) dominates, which could result in a later emergence point in LM pretraining. However, we emphasize that this explanation is only a plausible account of the observed trend: it remains unclear whether the later emergence at large context sizes observed in Figure 11 is indeed caused by retrieval difficulty, or whether this reversal is systematic across settings. If this account is correct, the predictive law presented in Section 6.1 may primarily describe the pre-saturation regime of effects (1) and (2), before increased retrieval difficulty in (3) begins to dominate. A precise characterization of this interplay remains an important future work.

## K. Analyses with other metrics

### K.1. Experiment 1

We replicate the analyses from Section 5 in other metrics: LA in Figure 12a and Figure 12b, AR (accuracy) in Figure 13a and Figure 13b, and AR (mean rank) in Figure 14a and Figure 14b. Notably, the shifting and slanting effects discussed in Section 5 are visibly captured by LA (Figure 12a) and by AR (mean rank; Figure 14a), but not as pronounced in AR (accuracy; Figure 13a). This is presumably, again, due to the discrete nature of accuracy, where the metric only captures whether the model ranks the target token as the most probable token or not. Another point worth noting is that, AR (accuracy) gives a false impression that a batch size of 512 fails in the associative recall task, as shown by the yellow curve in Figure 12a. However, when looking at the mean rank as opposed to accuracy, we can see that it goes through an abrupt drop (improvement) in the model trained with a batch size of 512 (Figure 14a), highlighting the importance of analyzing multiple metrics.

**(a)**

**(b)**

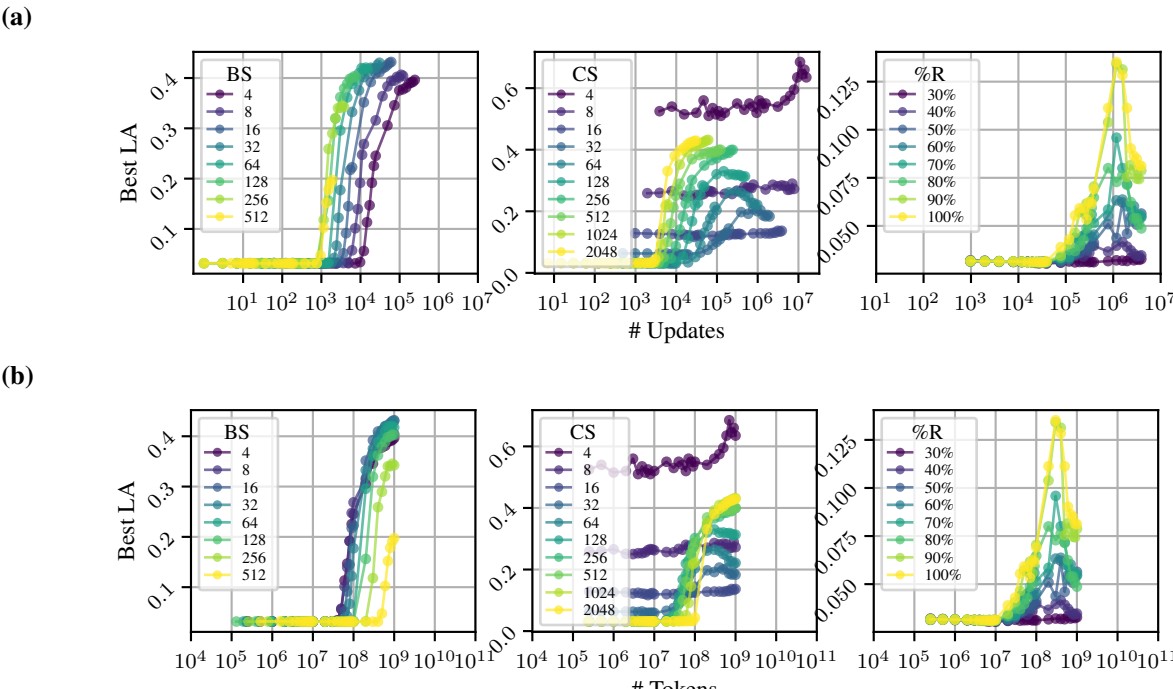

*Figure 12.* Developmental trajectories of LA of LMs with various batch sizes (left), context sizes (center), and repetitions (right) over the course of 1B tokens of pretraining, plotted against the number of updates (a) and the number of tokens (b). BS, CS, %R stand for batch size, context size, and the proportion of chunks with natural bigram repetitions, respectively.

**(a)**

**(b)**

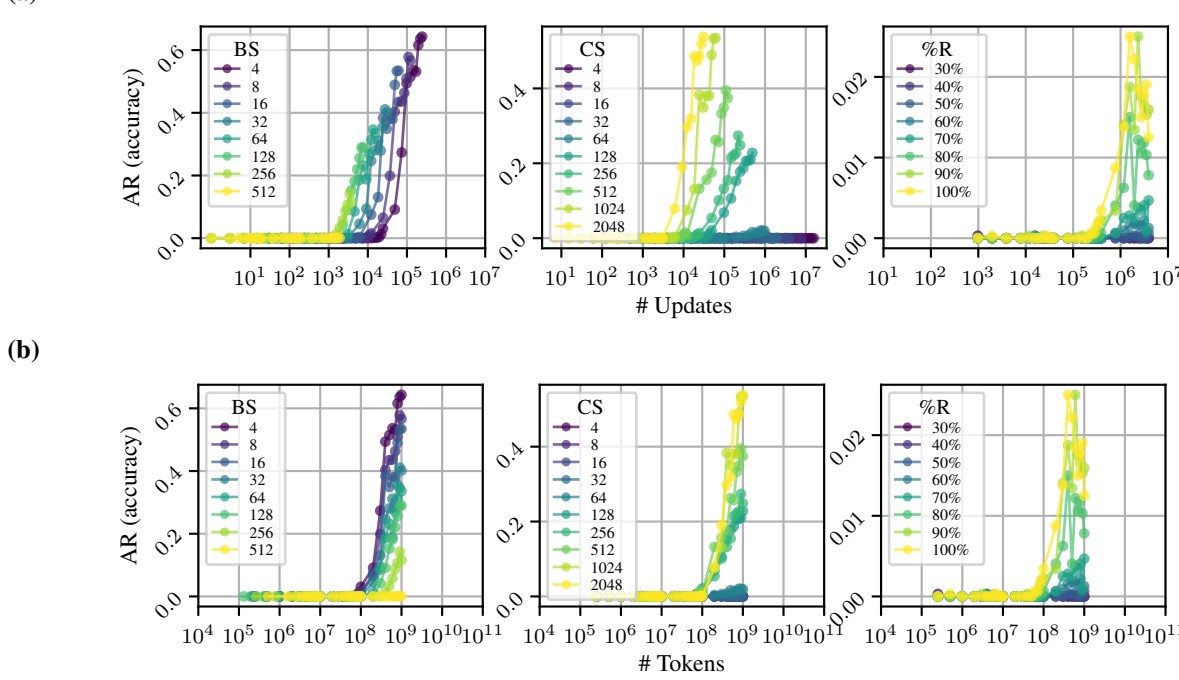

*Figure 13.* Developmental trajectories of AR (accuracy) of LMs with various batch sizes (left), context sizes (center), and repetitions (right) over the course of 1B tokens of pretraining, plotted against the number of updates (a) and the number of tokens (b). BS, CS, %R stand for batch size, context size, and the proportion of chunks with natural bigram repetitions, respectively.

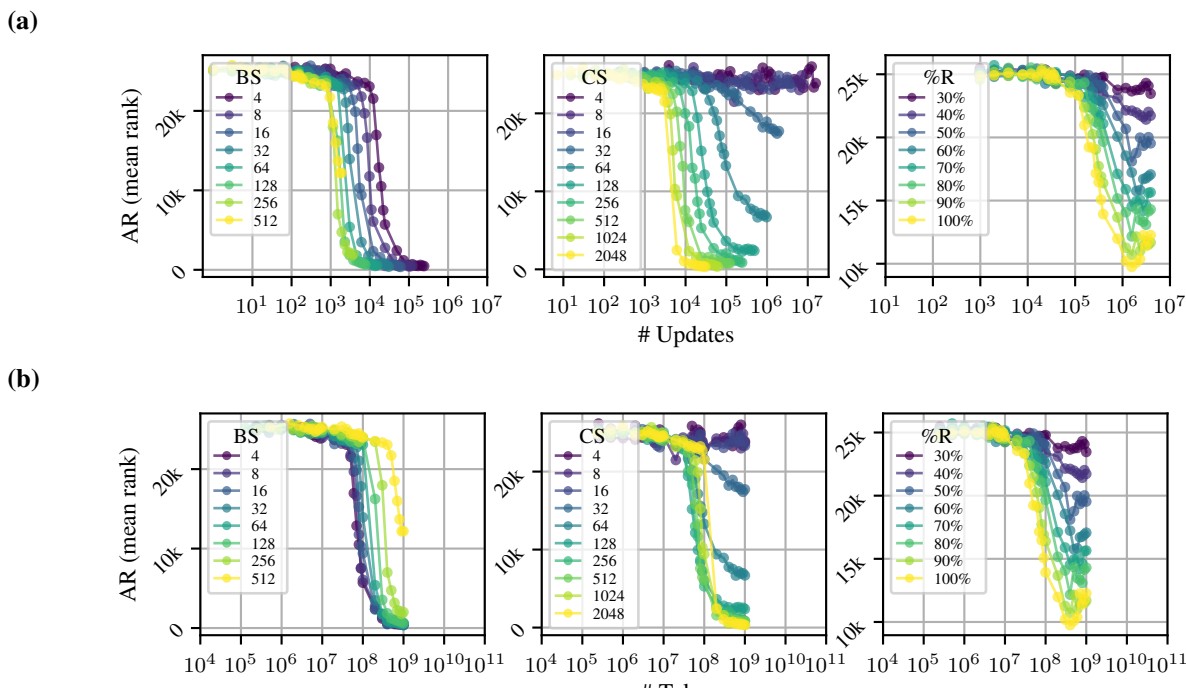

*Figure 14.* Developmental trajectories of AR (mean rank) of LMs with various batch sizes (left), context sizes (center), and repetitions (right) over the course of 1B tokens of pretraining, plotted against the number of updates (a) and the number of tokens (b). BS, CS, %R stand for batch size, context size, and the proportion of chunks with natural bigram repetitions, respectively.

### K.2. Experiment 2

We replicate the analyses from Section 7 with LA in Figure 15b, AR (accuracy) in Figure 16a, and AR (mean rank) in Figure 16b. Overall, the decision boundaries found in these results are consistent across different metrics. For example, PS and LA seem to draw the same line between the configurations where IHs emerge and those where IHs do not emerge. However, it is important to note that, while the bottom right region, $\in \{P(A, B, \ldots, A) > 0.1 \cap P(A, B, \ldots, A, B) > 0.5\}$, is filled with highly strong IHs (as indicated by bright yellow) for PS, it is not the case for LA. In particular, the bottom right configuration, $P(A, B, \ldots, A) = P(A, B, \ldots, A, B) = 0.9$, results in a moderate IH, weaker than other configurations with less repetitions.

AR (accuracy), on the other hand, seems not to show a line as clear as the other three metrics. While the no-IH region seems to be very similar, IH-regions near the decision boundary in PS and LA are more graded for AR (accuracy). This indicates that even with the emergence of strong IHs high in both PS and LA, AR accuracy can still be low. This again highlights the sequential nature of the emergence of latent and surface capabilities (Reddy, 2024).

**(a)**                                                                      **(b)**

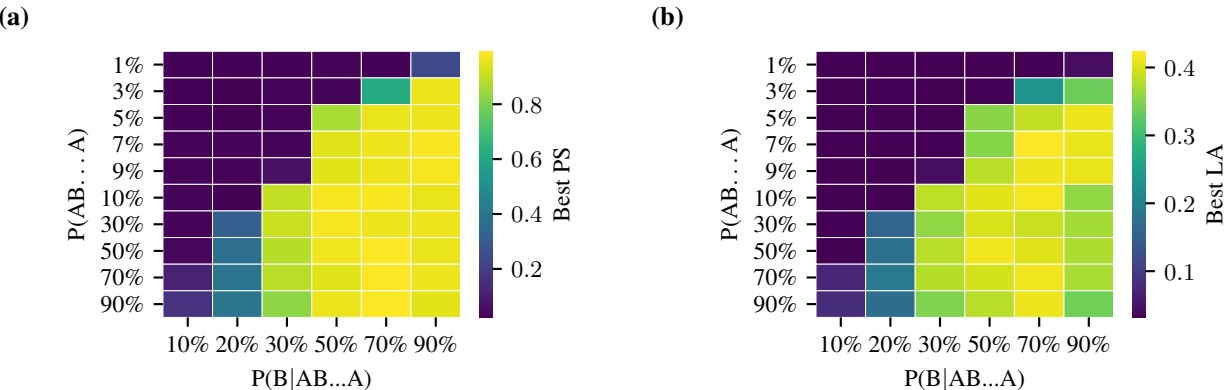

*Figure 15.* Best PS (a) and best LA (b) across all heads at the end of the training for each frequency reliability combination. Scores are represented in colors, with brighter colors representing higher scores.

**(a)**                                                                      **(b)**

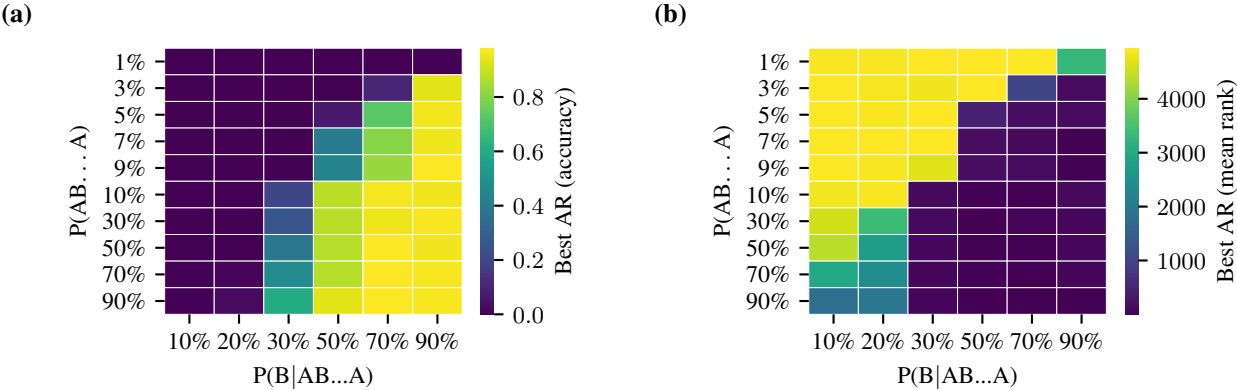

*Figure 16.* AR accuracy (a) and AR mean rank (b) at the end of the training for each frequency reliability combination. Scores are represented in colors, with brighter colors representing higher scores.

### K.3. Experiment 3

We showed in Section 8 that, when measured by PS, IHs form when the underlying distribution of the data generation process is +D and high in both repetition frequency and reliability (0.9-0.9) regardless of the distribution shape, and when it is +D+C and Zipfian when the repetition frequency and reliability are near the decision boundary (0.1-0.3). Here, we replicate the same analyses in LA (Figure 17b), AR (accuracy; Figure 18a), and AR (mean rank; Figure 18b). We make two observations. First, in each metric, Zipf$_{[+D+C]}$ is consistently the only distribution shape that promotes the emergence of IHs when the bigram repetition is near the decision boundary (0.1-0.3).

Second, and more importantly, contrary to the observation we made in Section 8, IHs do form even when the underlying distribution is $-D$, meaning that each token is i.i.d. However, it is important to note that the training data sample generated from this distribution is highly unlikely to be $-D$, and the tokens in the training data are not i.i.d., given the constrained generation process described in Appendix G. Because we impose $P(A, B, \ldots, A)$ and $P(A, B, \ldots, A, B)$ while sampling from the generation, samples are not independent of one another, violating the $-D$ constraint. However, the reason why these training data sampled from underlying distributions $*_{[-D-C]}$ with high bigram repetition (0.9-0.9) lead to a low score in PS (Figure 6) but high scores in other metrics (Figures 17b, 18a and 18b), remains unclear, and warrants further investigation. Generally, studying these heads that are low in PS but high in LA could potentially reveal an alternative mechanism for IHs.

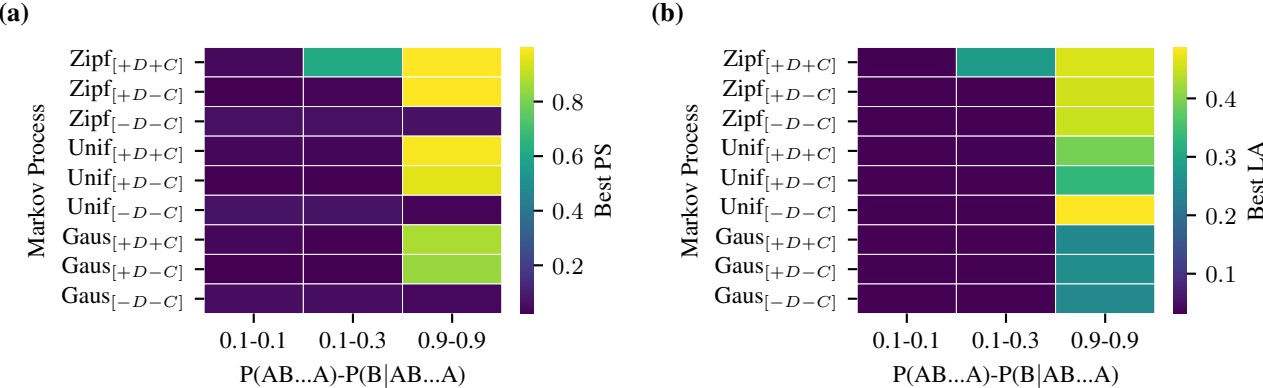

*Figure 17.* Best PS (a) and best LA (b) at the end of the training for each frequency reliability combination. Scores are represented in colors, with brighter colors representing higher scores.

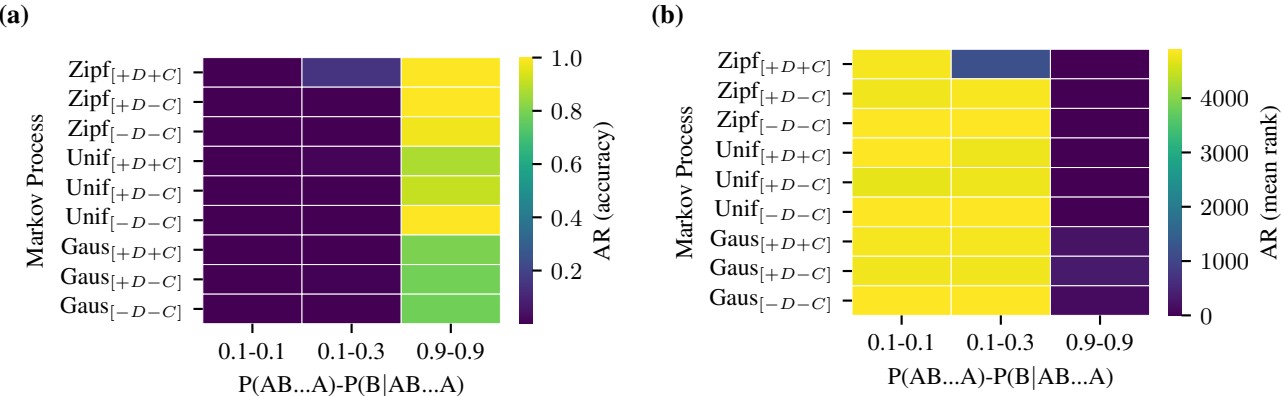

*Figure 18.* AR accuracy (a) and AR mean rank (b) at the end of the training for each frequency reliability combination. Scores are represented in colors, with brighter colors representing higher scores.

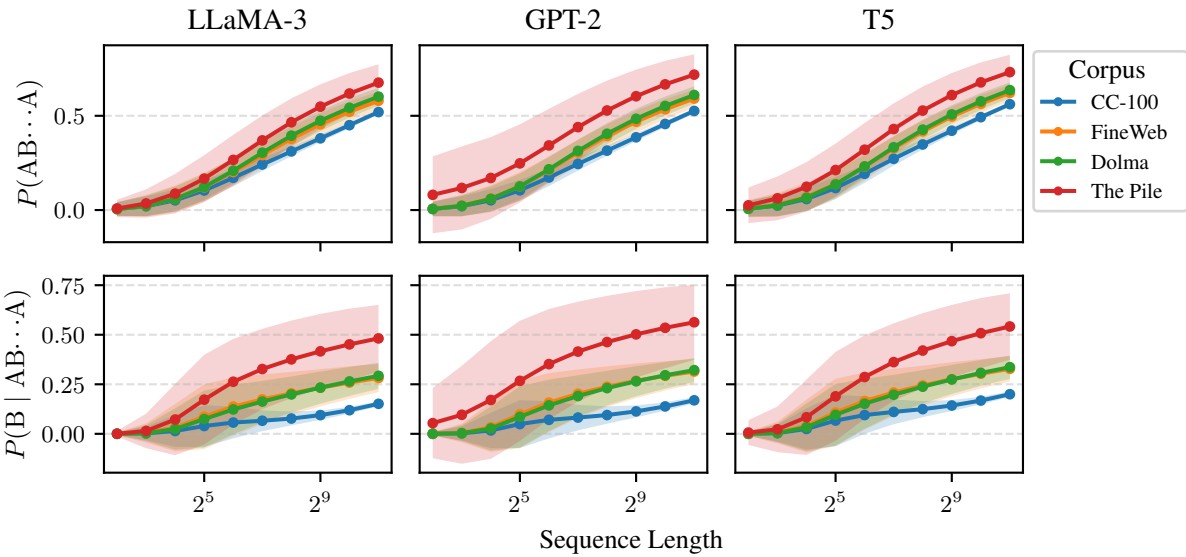

*Figure 19.* Effect of the choice of dataset on frequency $P(A, B, \ldots, A)$ (top) and reliability $P(B \mid A, B, \ldots, A)$ (bottom). Each column represents the tokenizer used to tokenize the dataset. The shaded region represents the standard deviation across sequences of the size indicated on the $x$-axis.

## L. Effect of Dataset and Tokenizer

In the main experiments, we used the pretrained GPT2 tokenizer (Radford et al., 2019) and the CC100 dataset (Conneau et al., 2020). A natural question is whether or not the reported results hold if the models are trained on a different dataset and/or a dataset tokenized by a different tokenizer. Since replicating the series of experiments with various combinations of datasets and tokenizers is beyond our compute budget, as a proxy experiment, we report frequency $P(A, B, \ldots, A)$ and reliability $P(B \mid A, B, \ldots, A)$ of various datasets with different tokenizers. Our datasets include CC100 (Conneau et al., 2020), The Pile (Gao et al., 2021), Dolma (Soldaini et al., 2024), and FineWeb (Penedo et al., 2024), and our tokenizers include GPT2 (Radford et al., 2019), LLaMA3 (Grattafiori et al., 2024), and T5 (Raffel et al., 2023).

Figure 19 summarizes, for each tokenizer, the effect of the choice of corpus on the 2 metrics: frequency and reliability. Across different tokenizers and sequence lengths, both frequency and reliability largely follow a consistent pattern: Pile > Dolma > FineWeb > CC-100. Although this warrants a more thorough investigation, a quick observation is that the web-crawl corpus (CC-100 and FineWeb) seems to have fewer repetitions than mixed domain corpora (Dolma and Pile). Given that the mean sequence length is Dolma > Pile > CC-100 > FineWeb, we surmise that this could be due to a domain effect rather than a simple document length effect. However, given that the Pile corpus has a much wider spread, and that Dolma and FineWeb are virtually indistinguishable in the 2 metrics, these observations should be taken with caution.

Figure 20 similarly summarizes the effect of tokenizer on frequency and reliability for each of the corpora. With the exception of the Pile corpus, across different corpora and sequence lengths, frequency $P(A, B, \ldots, A)$ seems to follow a consistent pattern: T5 > GPT2 > LLaMA-3. This is rather intuitive; T5, GPT2, and LLaMA-3 have vocabulary sizes of 32128, 50275, and 128256, respectively. The smaller the vocabulary size, the more likely a given word is tokenized into multiple tokens, resulting in more repetitions in a given sequence. Note that the shaded regions in Figure 19 and Figure 20 are based on the standard deviations over the sequences, and with a large enough sample size (> 10M tokens) used in this analysis, the standard error is so small that it would be invisible in the figures. Hence, the differences across corpora and tokenizers seem robust, and further investigation is needed to fully understand their effect on the repetitiveness of the training data, and more importantly, the emergence of IHs as well as phase transition in general.

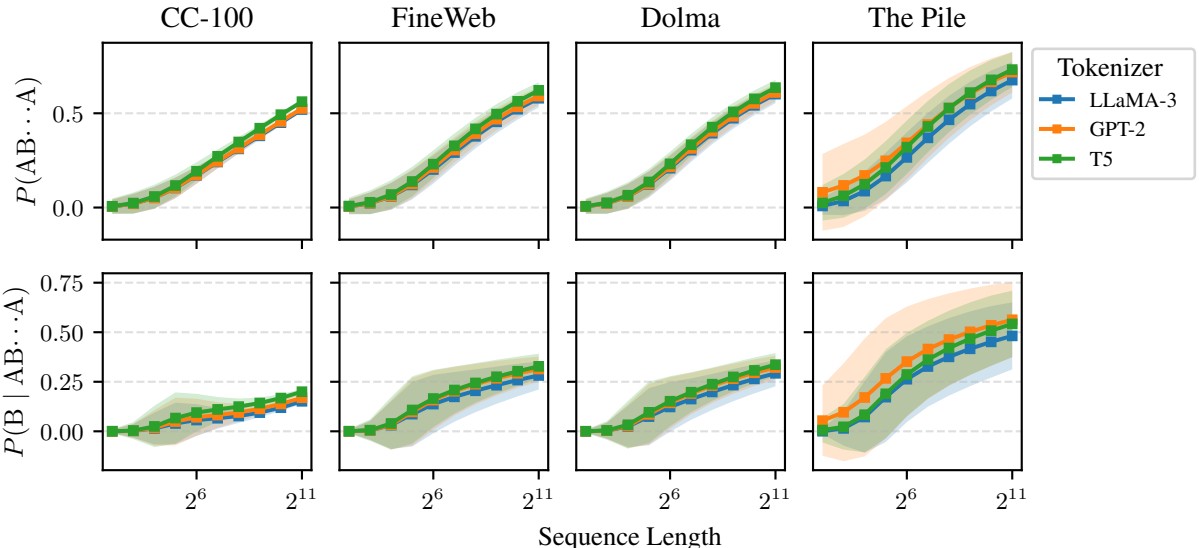

*Figure 20.* Effect of the choice of tokenizer on frequency $P(\mathsf{A}, \mathsf{B}, \dots, \mathsf{A})$ (top) and reliability $P(\mathsf{B} \mid \mathsf{A}, \mathsf{B}, \dots, \mathsf{A})$ (bottom). Each column represents the dataset. The shaded region represents the standard deviation across sequences of the size indicated on the $x$-axis.

*Table 8.* The full statistics of the effect of sequence length, corpus, and tokenizer on frequency and reliability. $P_A$ and $P_B$ refer to $P(\mathsf{A}, \mathsf{B}, \dots, \mathsf{A})$ and $P(\mathsf{B} \mid \mathsf{A}, \mathsf{B}, \dots, \mathsf{A})$, respectively. $\mu$ and $\sigma$ are sample mean and standard deviation across sequences, respectively.

| Dataset | Tokenizer | | | Sequence Length | | | | | | | | | |
|---|---|---|---|---|---|---|---|---|---|---|---|---|---|
| | | | | 4 | 8 | 16 | 32 | 64 | 128 | 256 | 512 | 1024 | 2048 |
| CC-100 | LLaMA-3 | $P_A$ | $\mu$ | 0.0052 | 0.0203 | 0.0516 | 0.1038 | 0.1709 | 0.2416 | 0.3111 | 0.3804 | 0.4503 | 0.5198 |
| | | | $\sigma$ | 0.0363 | 0.0519 | 0.0580 | 0.0544 | 0.0451 | 0.0342 | 0.0257 | 0.0199 | 0.0158 | 0.0122 |
| | | $P_B$ | $\mu$ | 0.0001 | 0.0023 | 0.0142 | 0.0395 | 0.0569 | 0.0660 | 0.0771 | 0.0943 | 0.1192 | 0.1514 |
| | | | $\sigma$ | 0.0083 | 0.0343 | 0.0801 | 0.1052 | 0.0801 | 0.0525 | 0.0348 | 0.0239 | 0.0172 | 0.0126 |
| | GPT-2 | $P_A$ | $\mu$ | 0.0058 | 0.0205 | 0.0519 | 0.1048 | 0.1728 | 0.2445 | 0.3153 | 0.3859 | 0.4565 | 0.5257 |
| | | | $\sigma$ | 0.0386 | 0.0528 | 0.0590 | 0.0554 | 0.0458 | 0.0350 | 0.0265 | 0.0207 | 0.0163 | 0.0125 |
| | | $P_B$ | $\mu$ | 0.0002 | 0.0024 | 0.0171 | 0.0492 | 0.0706 | 0.0820 | 0.0947 | 0.1133 | 0.1382 | 0.1691 |
| | | | $\sigma$ | 0.0107 | 0.0357 | 0.0896 | 0.1206 | 0.0938 | 0.0644 | 0.0439 | 0.0304 | 0.0215 | 0.0155 |
| | T5 | $P_A$ | $\mu$ | 0.0068 | 0.0237 | 0.0586 | 0.1167 | 0.1924 | 0.2718 | 0.3478 | 0.4209 | 0.4926 | 0.5616 |
| | | | $\sigma$ | 0.0415 | 0.0563 | 0.0625 | 0.0583 | 0.0480 | 0.0361 | 0.0266 | 0.0200 | 0.0154 | 0.0119 |
| | | $P_B$ | $\mu$ | 0.0002 | 0.0036 | 0.0247 | 0.0669 | 0.0946 | 0.1105 | 0.1246 | 0.1428 | 0.1678 | 0.1999 |
| | | | $\sigma$ | 0.0103 | 0.0429 | 0.1042 | 0.1295 | 0.0945 | 0.0624 | 0.0407 | 0.0276 | 0.0194 | 0.0141 |
| FineWeb | LLaMA-3 | $P_A$ | $\mu$ | 0.0054 | 0.0227 | 0.0591 | 0.1200 | 0.2018 | 0.2918 | 0.3778 | 0.4548 | 0.5216 | 0.5797 |
| | | | $\sigma$ | 0.0371 | 0.0572 | 0.0713 | 0.0769 | 0.0749 | 0.0693 | 0.0643 | 0.0602 | 0.0553 | 0.0488 |
| | | $P_B$ | $\mu$ | 0.0001 | 0.0046 | 0.0309 | 0.0866 | 0.1374 | 0.1739 | 0.2045 | 0.2326 | 0.2585 | 0.2817 |
| | | | $\sigma$ | 0.0084 | 0.0485 | 0.1197 | 0.1634 | 0.1477 | 0.1255 | 0.1081 | 0.0939 | 0.0823 | 0.0709 |
| | GPT-2 | $P_A$ | $\mu$ | 0.0062 | 0.0234 | 0.0607 | 0.1253 | 0.2121 | 0.3051 | 0.3920 | 0.4686 | 0.5344 | 0.5914 |
| | | | $\sigma$ | 0.0394 | 0.0579 | 0.0716 | 0.0761 | 0.0729 | 0.0673 | 0.0626 | 0.0585 | 0.0538 | 0.0479 |
| | | $P_B$ | $\mu$ | 0.0001 | 0.0044 | 0.0333 | 0.0981 | 0.1574 | 0.2023 | 0.2378 | 0.2676 | 0.2929 | 0.3143 |
| | | | $\sigma$ | 0.0086 | 0.0474 | 0.1247 | 0.1708 | 0.1491 | 0.1226 | 0.1028 | 0.0873 | 0.0739 | 0.0631 |
| | T5 | $P_A$ | $\mu$ | 0.0079 | 0.0279 | 0.0689 | 0.1381 | 0.2301 | 0.3274 | 0.4173 | 0.4957 | 0.5633 | 0.6218 |
| | | | $\sigma$ | 0.0450 | 0.0636 | 0.0760 | 0.0784 | 0.0731 | 0.0654 | 0.0583 | 0.0533 | 0.0484 | 0.0427 |
| | | $P_B$ | $\mu$ | 0.0003 | 0.0059 | 0.0396 | 0.1077 | 0.1655 | 0.2085 | 0.2441 | 0.2753 | 0.3031 | 0.3279 |
| | | | $\sigma$ | 0.0122 | 0.0546 | 0.1329 | 0.1699 | 0.1447 | 0.1191 | 0.1001 | 0.0858 | 0.0737 | 0.0639 |
| Dolma | LLaMA-3 | $P_A$ | $\mu$ | 0.0052 | 0.0222 | 0.0587 | 0.1221 | 0.2087 | 0.3044 | 0.3945 | 0.4739 | 0.5423 | 0.6016 |
| | | | $\sigma$ | 0.0361 | 0.0547 | 0.0659 | 0.0689 | 0.0663 | 0.0622 | 0.0585 | 0.0556 | 0.0520 | 0.0477 |
| | | $P_B$ | $\mu$ | 0.0001 | 0.0033 | 0.0242 | 0.0742 | 0.1230 | 0.1627 | 0.1987 | 0.2331 | 0.2645 | 0.2927 |
| | | | $\sigma$ | 0.0066 | 0.0408 | 0.1043 | 0.1439 | 0.1252 | 0.1046 | 0.0908 | 0.0812 | 0.0733 | 0.0658 |
| | GPT-2 | $P_A$ | $\mu$ | 0.0059 | 0.0224 | 0.0590 | 0.1253 | 0.2161 | 0.3138 | 0.4047 | 0.4841 | 0.5522 | 0.6105 |
| | | | $\sigma$ | 0.0382 | 0.0550 | 0.0662 | 0.0686 | 0.0648 | 0.0602 | 0.0563 | 0.0532 | 0.0500 | 0.0459 |
| | | $P_B$ | $\mu$ | 0.0001 | 0.0032 | 0.0268 | 0.0855 | 0.1436 | 0.1908 | 0.2307 | 0.2652 | 0.2955 | 0.3214 |
| | | | $\sigma$ | 0.0067 | 0.0400 | 0.1111 | 0.1544 | 0.1300 | 0.1047 | 0.0877 | 0.0759 | 0.0672 | 0.0603 |
| | T5 | $P_A$ | $\mu$ | 0.0069 | 0.0254 | 0.0653 | 0.1356 | 0.2315 | 0.3332 | 0.4265 | 0.5073 | 0.5766 | 0.6361 |
| | | | $\sigma$ | 0.0416 | 0.0585 | 0.0687 | 0.0694 | 0.0642 | 0.0577 | 0.0526 | 0.0486 | 0.0450 | 0.0410 |
| | | $P_B$ | $\mu$ | 0.0002 | 0.0043 | 0.0329 | 0.0958 | 0.1517 | 0.1974 | 0.2377 | 0.2744 | 0.3074 | 0.3363 |
| | | | $\sigma$ | 0.0091 | 0.0467 | 0.1205 | 0.1551 | 0.1264 | 0.1005 | 0.0841 | 0.0732 | 0.0651 | 0.0587 |
| The Pile | LLaMA-3 | $P_A$ | $\mu$ | 0.0083 | 0.0349 | 0.0872 | 0.1669 | 0.2660 | 0.3696 | 0.4656 | 0.5486 | 0.6183 | 0.6760 |
| | | | $\sigma$ | 0.0462 | 0.0752 | 0.1047 | 0.1242 | 0.1330 | 0.1323 | 0.1270 | 0.1185 | 0.1085 | 0.0973 |
| | | $P_B$ | $\mu$ | 0.0004 | 0.0147 | 0.0721 | 0.1724 | 0.2629 | 0.3270 | 0.3758 | 0.4164 | 0.4514 | 0.4816 |
| | | | $\sigma$ | 0.0145 | 0.0872 | 0.1801 | 0.2268 | 0.2164 | 0.2029 | 0.1956 | 0.1888 | 0.1806 | 0.1693 |
| | GPT-2 | $P_A$ | $\mu$ | 0.0807 | 0.1172 | 0.1702 | 0.2476 | 0.3429 | 0.4399 | 0.5281 | 0.6038 | 0.6668 | 0.7183 |
| | | | $\sigma$ | 0.2036 | 0.2202 | 0.2167 | 0.2071 | 0.1940 | 0.1767 | 0.1586 | 0.1413 | 0.1248 | 0.1082 |
| | | $P_B$ | $\mu$ | 0.0538 | 0.0961 | 0.1707 | 0.2672 | 0.3518 | 0.4150 | 0.4631 | 0.5019 | 0.5350 | 0.5629 |
| | | | $\sigma$ | 0.1775 | 0.2471 | 0.2971 | 0.3036 | 0.2778 | 0.2521 | 0.2333 | 0.2183 | 0.2045 | 0.1894 |
| | T5 | $P_A$ | $\mu$ | 0.0250 | 0.0621 | 0.1236 | 0.2122 | 0.3202 | 0.4297 | 0.5277 | 0.6097 | 0.6773 | 0.7317 |
| | | | $\sigma$ | 0.0942 | 0.1160 | 0.1312 | 0.1402 | 0.1415 | 0.1359 | 0.1275 | 0.1174 | 0.1059 | 0.0925 |
| | | $P_B$ | $\mu$ | 0.0063 | 0.0227 | 0.0838 | 0.1893 | 0.2867 | 0.3618 | 0.4203 | 0.4675 | 0.5079 | 0.5418 |
| | | | $\sigma$ | 0.0626 | 0.1152 | 0.1904 | 0.2243 | 0.2105 | 0.1958 | 0.1886 | 0.1831 | 0.1773 | 0.1677 |

