# OpenReview forum: "Predicting the Emergence of Induction Heads in Language Model Pretraining"
_ICML.cc/2026/Conference — ICML 2026 regular_

### Official Review · Reviewer_59F4 · 2026-02-22

**Soundness:** 3
**Presentation:** 3
**Significance:** 4
**Originality:** 3
**Overall Recommendation:** 4
**Confidence:** 4

**Summary:**

This paper investigates the emergence of induction heads (IHs) during pretraining, focusing on the question of when and why they form. The paper presents three main contributions: (1) a predictive law relating batch size and context size to the phase transition point where IHs emerge, (2) a characterization of bigram repetition frequency and reliability as key factors affecting IH formation, with a Pareto frontier identified in this two-dimensional space, and (3) an analysis using synthetic data showing that local dependency is necessary but not sufficient, and that categoriality and Zipfian marginal distributions matter near the Pareto frontier. The paper uses primarily 2-layer GPT2 models trained on CC100 English.

**Compliance With Llm Reviewing Policy:**

Affirmed.

**Final Justification:**

The authors were responsive throughout the discussion, providing new analyses on utility of predicting IH emergence, seed-level variance, and tokenizer,corpus robustness. These additions, along with the promised revisions addressed most of my concerns. I raised my score from 3 to 4. The paper is interesting and asks a research question of interest to the ICML audience.

**Key Questions For Authors:**

Do models where IHs emerge earlier end up meaningfully different after full training (in loss, ICL accuracy, or any other measure)? Is there path dependence, or just a timing shift? I feel this (+ accompanied by some writing changes) would be the most significant to me in updating my rating.

How sensitive is T = 750,000 to corpus and tokenizer?

Can you provide variance estimates across seeds for representative conditions?

What is the functional form of the Pareto frontier?

**Limitations:**

Yes

**Strengths And Weaknesses:**

**Strengths**
1. Induction heads are an important setting/object of research, and understanding their formation is a worthwhile goal. I do agree with the claim that IHs have been implicated as a core mechanism underlying in-context learning (although I would point out that recent work by Sahin et al. 2025 may complicate this somewhat), and their abrupt emergence during pretraining is one of the most striking and reproducible phenomena in training dynamics research.

Kerem Sahin, Sheridan Feucht, Adam Belfki, Jannik Brinkmann, Aaron Mueller, David Bau, Chris Wendler. "In-Context Learning Without Copying." arXiv preprint arXiv:2511.05743, (2025).

2. Both the predictive law and the cross-validation thereof are strong results. I appreciated the thoroughness of the validation.

3. I thought that breaking down "burstiness" (Chan et al., 2022) into P(A,B,..,A) and P(B|A,B,...,A) was an improvement over prior formulations. I think the finding that reliability was more than raw frequency for IH formation is of interest to practitioners and the broader research community.

Chan, S. C., Santoro, A., Lampinen, A. K., Wang, J. X., Singh, A. K., Richemond, P. H., McClelland, J., and Hill, F. Data distributional properties drive emergent incontext learning in transformers. In Oh, A. H., Agarwal, A., Belgrave, D., and Cho, K. (eds.), Advances in Neural Information Processing Systems, 2022. URL https://openreview.net/forum?id=lHj-q9BSRjF.

4. In general, I appreciated the experimental design (there's a nice, natural flow of experiments from natural to semi-natural to synthetic data) and the thoroughness of metrics (and their motivation).

**Weaknesses**

1. I would like to hear more about what to do with these findings. The paper never really establishes why predicting IH emergence timing matters, as opposed to just being happy to predict it happens at all. There are many possible routes to take here; I list a few (but am not really opinionated about which one seems best).
* **Training efficiency.** Does IH emergence correlate with faster convergence or better final loss? If so, the scaling law could be a practical tool for selecting batch/context size schedules.
* **Hysteresis / path dependence.** The paper seems to implicitly assume something like: the point at which IHs emerge matters because it shapes subsequent learning dynamics---models that form IHs earlier might develop qualitatively different downstream capabilities. This is a testable and interesting hypothesis (and would connect to work on curriculum effects and phase-ordering in training), but it is never articulated, let alone tested. Do models that form IHs at 10M tokens end up meaningfully different from those that form them at 500M tokens, beyond the timing shift itself? And we may be interested further in how this affects other downstream properties of the model, in the more general competencies language models are famous for (such as overall fluency, effective context window, question answering capabilities on various domains, etc.).
* Conversely, one might want to delay or prevent IH formation, and knowing the timing of their emergence may inform such efforts (as per Sahin et al. 2025 above).

2. The paper is hard to read at times. There are some low hanging fruit that should be easy to fix:
* Core quantities are defined in appendices. P(A,B,...,A) and P(B|A,B,...,A) are central to two of three experiments but aren't properly defined until Appendix F (page 15). The main text provides informal glosses insufficient for understanding the experimental design.
* Imprecise coined terms do heavy lifting. Some load-bearing terms like "shifting effect" and "slanting effect" are used repeatedly but never defined. What exactly constitutes each? The reader must reverse-engineer the meaning from figures, which are quite cluttered and hard to read.
* Section 5.2 is particularly hard to parse. It jumps between batch size, context size, and repetition effects with forward references to appendices for important experimental details. Ordinarily I am a fan of large appendices, but in this case i have no way of knowing of some of these details are load-bearing for the experiments and the interpretation of the results.

3. I find the support for the conclusions of experiment 3 the weakest. First, the factorial design is compromised by generation artifacts; Algorithm 1 imposes constraints only on sequence second halves, and under −D conditions, the generation itself violates independence (which the authors point out in Section J.3). The appendix then reveals that IHs do form under −D with high repetition when measured by LA and AR, which contradicts the main text's claim that "local dependency is a necessary condition." Most importantly, the key interaction rests on one data point. The claim that Zipf[+D+C] uniquely promotes IH formation near the Pareto frontier comes from a single column of Figure 5 (0.1-0.3), with 9 conditions tested. I think, scientifically, it would be hard to conclude a three-way interaction from this alone (e.g., no error bars, confidence intervals, repeated seeds, or even bootstrapping appear anywhere. Emergence point identification via piecewise linear fitting is inherently noisy. For a paper fitting regression models to emergence points and drawing decision boundaries, this is quite an omission!). I'd also guess that for a Pareto frontier, a 5×10 grid is too coarse near the boundary to locate it. (It is also not a Pareto frontier in the standard multi-objective optimization sense---it is a decision boundary.)

Some minor errors that did not factor significantly into my rating, but the authors would do well to improve on!
* "creating creating |C| groups" — duplicated word (line 360)
* P used for both probability and model size (Eq. 5 vs. everywhere else); Table 3 uses S for the same quantity
* Table 1: −D−C rows show "−" for similarities that are 1 by definition
* "%NR" = proportion with no repetitions, forcing double-negative reasoning; in my opinion reporting the positive would be clearer
* Figure 3 caption reports r = .98; text says r > .99 pre-log and .98 post-log; figure is in log space. which is one is being shown?
* Algorithm 1 line 7: no token repeats in the first half, a strong undiscussed constraint

Overall, I find the paper asks a significant, well-scoped question and provides a clean scaling law (Experiment 1) and a useful conceptual decomposition of repetition (Experiment 2). But I do feel it suffers from an issue mainly of motivation: it characterizes IH emergence in impressive detail without establishing why the characterization of the timing is important to us. The paper appears to assume that the timing of IH emergence has downstream consequences, but this is never tested or even stated as a hypothesis. My other large issue with the paper is with the support experiment 3 can provide its conclusions, mainly being exploratory or too coarse to conclude strongly in either direction. And of course, the lack of uncertainty quantification is hard to overlook when fitting such scaling laws.

---

> ### Author Rebuttal · Authors · 2026-03-30
>
> Thank you for taking the time to review our paper, and for providing insightful feedback!
> 1. **Practical implication**
> * We agree that this should be made more explicit. In follow-up analysis, we found evidence for a practical training efficiency interpretation. We evaluated all 50M models (with context size $\\ge 64$) on a val set of 64-token sequences, so that models with larger context sizes do not gain an unfair advantage from additional test-time context. Under this controlled comparison, we observe a clear trend: models whose IHs emerge earlier in the number of updates tend to achieve lower final val loss at the end of training (1B tokens). This suggests that IH emergence timing is not merely descriptive, but predictive of training outcomes under a fixed token budget.
> * One exception occurs for models with batch size $>128$. Although these models exhibit earlier IH emergence, their final val loss is worse than that of smaller-batch models. We suspect this reflects the critical batch size tradeoff (McCandlish et al., 2018): increasing batch size can improve gradient quality and accelerate some aspects of learning, but beyond the critical batch size, the reduction in the number of parameter updates hurts final performance. In other words, IH emergence timing and final loss are related, but they are not governed by exactly the same optimization tradeoff. This trend was consistent across 3 random seeds.
> * We agree that the path-dependence claim would be very interesting; however, this would require substantially larger-scale and longer training runs, which give a more precise estimate of final convergence. At our current scale, we do not think we can yet make a convincing claim about asymptotic behavioral differences. We will therefore narrow our claim accordingly: our evidence supports the practical relevance of IH emergence timing for training efficiency, while the path-dependence claims remain open for future work. More broadly, emergence timing may also matter because it can potentially be manipulated. For example, Chen et al. (2024) show that briefly suppressing the emergence of a related specialized attention structure can improve training outcomes. This suggests that predicting phase transitions is valuable for intervention during training.
> 2. **Writing**
> * We will make sure to fix the noted typos/inconsistencies. With the additional page, we should be able to include a more precise definition of P(AB…A) and P(B|AB…A) in the main text. As to shifting and slanting, we will also define them more rigorously, e.g., as slopes of fit lines.
> * As to Section 5.2, we tried our best to discuss in the order of batch size, context size, and repetition, both in the method and result sections. Is the parsing difficulty mostly coming from the predictive law section, with several forward references to the appendices? Do you have a recommendation as to which ones are particularly hard to read/benefit from including in the main body? We are happy to incorporate any concrete suggestions into the CR version!
> 3. **Uncertainty (including the constant T)**
> * For the predictive law (experiment 1), we have added 6 models with a selected subset of 6 configs, each with 2 random seeds (so we can quantify the uncertainty across 3 random seeds for each config), and the emergence points appear very robust:
> |batch-context|log10(Emergence point)|
> |-|-|
> |16-64|4.308±0.060|
> |16-256|3.828±0.064|
> |16-1024|3.507±0.029|
> |16-2048|3.458±0.008|
> |128-1024|2.987±0.023|
> |512-1024|2.910±0.034|
> * We also fit a predictive law on these 6 models and show that, together with the CV results, the variance is reasonable across different seeds:
> |Parameter|Value|
> |-|-|
> |$\\log(T)$|13.540±0.401|
> |$\\beta$|-0.470±0.026|
> |$\\gamma$|-0.589±0.048|
> |$R^2$|0.968±0.005|
> * And finally, for Experiment 3, we additionally trained 3 configs (P(AB…A)=0.1, P(B|AB…A)=0.3, [+D+C], for each of the 3 distributions) with 2 random seeds, and show that for all 3 seeds, the conclusion remains the same (i.e. only [+D+C] with Zipfian distribution leads to IH emergence). We agree that the sampling procedure affects the distribution of the sampled datasets, and that this is especially true for strong constraints such as P(AB…A)=0.9, P(B|AB…A)=0.9. We will include this in the main text.
> 4. **Functional form of the decision boundary:**
> * We first directly model the PS for each P(AB…A), P(B|AB…A) combinations: $\\hat{PS}=\\sigma(k(\\alpha\\log p_a+\\beta\\log p_b-\\tau))$, where $p_a$ and $p_b$ are P(AB…A), P(B|AB…A), respectively. Fitting this to our data, we obtain $\\alpha=0.5674, \\beta=1.1531, \\tau = -2.6407, k = 9.5950$. Plugging in these values and solving for $p_b$ for when $\\hat{PS}=0.5$, we obtain $p_b = e^{\\tau / \\beta} pa^{-\\alpha/\\beta}\\approx 0.1013 \\times pa^{-0.4920}$. The predictive form fits our observation that PS is more sensitive to P(B|AB…A) than it is to P(AB…A). Also, we agree that "decision boundary" is more appropriate, and will revise accordingly.

---

> > ### Author Rebuttal · Reviewer_59F4 · 2026-04-01
> >
> > Thanks for the response! While a few of my concerns are only partially resolved, I am already happy to raise my score from 3 (Weak Reject) to 4 (Weak Accept) contingent on the authors including the val loss and variance results in the revised manuscript. Point by point:
> >
> > 1. I am happy with the new analysis showing that earlier IH emergence (in updates) correlates with lower final validation loss. I think the critical batch size is a reasonable explanation, and I do think it is fine to leave something as large as hysteresis for future work. The connection to Chen et al. (2024), where suppressing phase transitions can improve training outcomes, further strengthens the case that predicting emergence timing has practical value. I consider this concern substantively addressed and I would encourage the authors to feature this result prominently in the revision!
> >
> > 2. Thank you very much for the seed-level variance estimates, these are helpful and do suggest the predictive law is robust! The fitted parameters also appear stable across seeds, so I consider this concern addressed.
> >
> > 3. Thank you for fitting the linear model and for agreeing to adopt "decision boundary" over "Pareto frontier." The finding that the coefficient on reliability is more than twice that on frequency is a nice result.
> >
> > 4. I note that this question of sensitivity of T to tokenizer and corpus was not addressed in the rebuttal. The constant T is the central quantity of the predictive law, and understanding its generality is important. If T shifts substantially under a different corpus (e.g., code, multilingual data) or tokenizer (e.g., BPE with a different vocabulary size, SentencePiece, character-level), the law's practical utility becomes corpus-specific rather than general. Even a brief discussion of why T might or might not be expected to generalize, or a preliminary experiment with one alternative corpus or tokenizer, would help. I would appreciate the authors' thoughts on this, and I would ask that the revised manuscript include at least a discussion paragraph on this.
> >
> > 5. I appreciate the 3-seed replication of the key condition in experiment 3, which does strengthen the Zipfian finding. However, two of my original concerns remain only partially addressed:
> >
> > The -D contradiction across metrics. The rebuttal does not engage with the fact that IHs do form under -D conditions when measured by LA and AR (as the authors themselves report in Appendix J.3), which complicates the main text's claim that "local dependency is a necessary condition." The authors acknowledge in the appendix that the constrained generation process violates the -D assumption in the sampled data, and that the discrepancy across metrics "remains unclear and warrants further investigation." I think the main text should reflect this nuance rather than stating the necessity of local dependency as a straightforward conclusion.
> >
> > Coarseness of the grid near the boundary. The 3-seed replication helps with the specific (0.1, 0.3) cell, but the broader concern stands: the interaction claim bears on a narrow slice of the space. Another one or two additional grid points near the boundary (e.g., (0.1, 0.2) or (0.05, 0.3)) would help readers (and myself) assess how sharp the transition is.
> >
> > 6. Finally, on writing. Thank you, such fixes would be great! About Section 5.2: the parsing difficulty stems primarily from the fact that the "shifting" and "slanting" effects are introduced as observations without formal definitions, and that the reader must hold several forward references in mind simultaneously (Appendix E for repetition manipulation, Appendix C for the inverse shifting effect, Appendix F for frequency/reliability). My concrete suggestion would be: (a) define "shifting" and "slanting" explicitly at first use, and (b) move the key content of Appendix E (how repetition rate is controlled) into the main text, since it is necessary for interpreting Figure 2 (right). Or, at least as a footnote.

---

> > > ### Author Response · Authors · 2026-04-02
> > >
> > > Thank you for your continued engagement in the discussion!
> > >
> > > * **1, 2, 3**: We are glad that our clarifications and additional experiments addressed your concerns. And yes, we will be sure to reflect them in our final draft, as well as highlighting the uncertainty estimation experiments!
> > > * **4 (robustness of $T$ across tokenizers and corpora)**: We are sorry that we have missed your point on the variance introduced by the tokenizer/corpus. We would like to summarize a few follow-up analyses we have newly conducted, as well as why we think the constant should be robust.
> > >     * We think a tokenizer of different kinds (e.g. word-level tokenizer as opposed to subword tokenizer) could affect the emergence of induction heads (including the constant $T$), because the tokenization of long words into multiple tokens increases the rate of bigram repetitions (e.g., “encoders and decoders” → [enc, **od, ers,** and, dec, **od, ers**]). Yet, given that most modern models use a variant of subword tokenization, we believe that in the context of modern LLMs, the choice of a different tokenizer should not largely affect the observed results.
> > >      * As a proxy experiment, we measured how $P(AB…A)$ and $P(B|AB…A)$ vary for different tokenizers when corpus is fixed, and for different corpora when tokenizer is fixed, for each of the log-spaced context sizes (4-2048). We tested 3 pretrained tokenizers: T5 (sentencepiece unigram), Llama3 (more recent byte-level BPE), on top of GPT2 (byte-level BPE), as well as 4 corpora: Dolma, Fineweb, and Pile, on top of CC100. When varying the tokenizer while fixing the corpus, repetitions were T5 > GPT2 > Llama3, consistently across corpora. We speculate that this is due to the difference in vocabulary size, where a smaller vocabulary size leads to a finer-grained tokenization, increasing the number of repetitions. Indeed, T5, GPT2, and Llama3 have roughly 30k, 50k, and 120k vocabulary sizes, respectively. When varying the corpus while fixing the tokenizer, repetitions were Pile <= Dolma < Fineweb <= CC100. It appears that mixed-domain corpora (Pile and Dolma) seem to have fewer repetitions than single-domain web corpora (Fineweb and CC100), although a more thorough investigation lies outside the scope of our study.
> > >     * With that said, the cross-fold validation results of the predictive law in Appendix D.3 (p.14) are fairly robust, and this includes both GPT2 models (trained on CC100, with GPT2 tokenizer) and Pythia models (trained on Pile, with GPT-NeoX tokenizer). We noticed that we were not reporting the CV results for the constant $T$ (or the $\\alpha$ in $e^{\\alpha}$), so we are reporting it below (this is without the model size, so the right half of Table 5, p.14):
> > > |fold|alpha|beta|gamma|r2_test|
> > > |-|-|-:|-:|-:|
> > > |0|**13.297**|-0.362|-0.620|0.765|
> > > |1|**12.641**|-0.353|-0.539|0.897|
> > > |2|**13.277**|-0.360|-0.622|0.928|
> > > |3|**13.286**|-0.399|-0.602|0.834|
> > > |4|**13.530**|-0.381|-0.648|0.823|
> > >
> > >         This shows that $\\alpha$ has **a mean 13.206 and sd of 0.298**. We will be sure to include this discussion on the choice of tokenizer and corpus in the paper!
> > > * **5 ([-D] in Experiment 3)**: Yes, we agree that the [-D] condition is not as straightforward, and we will be sure to reflect that in the final draft.
> > > * **5 (Grid coarseness in Experiment 2)**: We suppose that the coarseness issue is mainly concerning Experiment 2. We are happy to add more data points if that helps - as you pointed out, the boundary seems to lie between the P(B|AB…A)=0.1 and P(B|AB…A)=0.3 columns, so adding more data points where P(B|AB…A)=0.2 could be informative. Is this a correct understanding of your suggestion?
> > > * **6 (Writing)**: Thank you for the concrete suggestions - with the additional page, we will be sure to add definitions for shifting and slanting, as well as bringing the repetition manipulation method to the main text!

---

### Official Review · Reviewer_CkL7 · 2026-03-06

**Soundness:** 4
**Presentation:** 4
**Significance:** 2
**Originality:** 2
**Overall Recommendation:** 4
**Confidence:** 4

**Summary:**

The paper presents an empirical analysis of induction head formation in LM pretraining. On natural language data, they propose a predictive law for when induction heads form; a main finding is that this law only depends on the batch size and context length, but is agnostic of the model size. They then study bigram repetition properties of the data based on the frequency of patterns of type <A, B, .., A>, and their reliability (fraction of times B is the next token) because these patterns capture the essence of that IHs do. In a semi-synthetic setup, where a Markov process is fitted from the data, they find a Pareto frontier, and further find that models have stronger sensitivity to reliability than to frequency. On fully synthetic data, they manipulate the dependency structure, categoricality and marginal distribution structure of the data and find that local dependency is a necessary condition, while marginal distribution and categoricality matters only close to the Pareto frontier of the bigram repetition structure.

**Compliance With Llm Reviewing Policy:**

Affirmed.

**Final Justification:**

The authors addressed my concerns. I maintain my opinion that this is a very well-executed paper, where the main weakness is the slightly limited novelty. I think my current score reflects this opinion well.

**Key Questions For Authors:**

I only have a small comment: I find the training plots a bit difficult to parse as the markers are too large. I recommend the authors decrease the markersize.

**Limitations:**

yes

**Strengths And Weaknesses:**

**Strengths**
- **Significance**
    - The paper studies the interesting and relevant problem of the mechanics of an IH head formation.
    - The model agnostic nature of IH formation is a very interesting result (but see limitation in 'Weaknesses')
- **Clarity:** The paper’s presentation is very clear and precise. The motivations are clear, and the paper is very easy to follow.
- **Soundness:** I find the methodology and experiment design very sound: experiments isolate factors of influence cleanly (e.g. the ‘shifting’ and ‘slanting’ effects or measuring IH formation timepoints  in terms of updates vs tokens in Section 5.2). The synthetic setups provide clear axes of manipulation, which are systematically tested.
 - **Very precise discussion of related work:** The paper discusses connections to prior works with great detail and accuracy, some examples include the connection to burstiness (addressed at length in Appendix F) and the hypothesized resolution of the conflict with Zucchet et al (2025) (section 8.1).

**Weaknesses**
- **Novelty:** Apart from the predictive law for IH formation, which is an impactful result, I find the paper’s ambition and results somewhat limited. The (semi)-synthetic setups test whether local dependency, categorically and the shape of the marginal distribution affect IH formation. Local dependency seems like a trivial condition, and the other two metrics have been studied already to some extent in the context of ICL (Chan et al, 2022.), as the authors note. Therefore, in terms of the paper’s goals, these sections read more as an extension (e.g. the frequency and reliability metrics make the ‘burstiness’ notion more precise) or simple adaptation of existing ideas to the IH setup than a novel contribution.
- **Significance**: The predictive law is only fitted / tested on Transformers, hence it is unclear whether the model agnosticness only holds within the Transformer class

Overall, the main strength of this paper is its quality of execution. Despite the limited novelty, the paper formulates clear, well-motivated hypotheses, tests them in controlled settings, and reaches clear findings. I believe that such systematic studies benefit the community, hence I recommend weak acceptance.

**References**

Chan, S. C., Santoro, A., Lampinen, A. K., Wang, J. X.,Singh, A. K., Richemond, P. H., McClelland, J., and Hill, F. Data distributional properties drive emergent in-context learning in transformers. NeurIPS, 2022. https://openreview.net/forum?id=lHj-q9BSRjF.

Zucchet, N., D’Angelo, F., Lampinen, A. K., and Chan, S. C. The emergence of sparse attention: impact of data distribution and benefits of repetition. NeurIPS 2025. URL https://openreview.net/forum?id=jMhRbV47pS.

---

> ### Author Rebuttal · Authors · 2026-03-30
>
> Thank you for taking the time to read our paper, and for providing the feedback!
> 1. **Significance**
> Induction heads have primarily been studied in Transformer language models, so we do not view our focus on Transformer LMs as a limitation of this work. Extending the analysis to non-Transformer architectures is an interesting direction for future research - we will make sure to clarify this scope as well as potential future directions in the main text!
> 2. **Plot presentation:**
> We will make the marker sizes smaller for better readability. Thank you for your suggestion!

---

> > ### Author Rebuttal · Reviewer_CkL7 · 2026-04-02
> >
> > Thank you for the rebuttal! I have looked through the other rebuttals as well. I maintain my opinion that this is a very well-executed paper, where the main weakness is the slightly limited novelty. I think my current score reflects this opinion well.

---

> > > ### Author Response · Authors · 2026-04-02
> > >
> > > Thank you again for your feedback and for acknowledging our rebuttal!

---

### Official Review · Reviewer_bmrH · 2026-03-13

**Soundness:** 2
**Presentation:** 2
**Significance:** 3
**Originality:** 3
**Overall Recommendation:** 5
**Confidence:** 2

**Summary:**

This paper first investigates the phase transition marking the emergence of induction heads during large language model (LLM) pretraining on natural language datasets. By pinpointing the exact moment of this transition using established metrics such as the prefix-matching score, the authors demonstrate that the emergence timing can be predicted with high accuracy as a function of batch size, context size, and model size. Subsequently, utilizing synthetic data, the study experimentally and quantitatively analyzes which specific properties of natural language are essential for the formation of these induction heads. Specifically, the authors define the concepts of "frequency" and "reliability" regarding bigram occurrences within a context. By manipulating these as control parameters, they experimentally demonstrate that the prerequisites for induction head formation constitute a Pareto frontier. Furthermore, the paper examines how other inherent linguistic properties, namely Zipf's law and word categoriality, influence the emergence of induction heads.

**Compliance With Llm Reviewing Policy:**

Affirmed.

**Final Justification:**

In my initial assessment, I pointed out a weakness in the paper's soundness, particularly regarding the lack of architectural diversity and the uniform application of the reliability parameter.However, the authors' highly constructive rebuttal successfully alleviated these main concerns. This was achieved by highlighting the validation results on the Pythia 7B models and, more importantly, by providing a detailed and compelling explanation regarding my concern about the potential impact of the artificially set conditional bigram probability ($P(B|A)$) threshold on the experimental outcomes.Weighing these factors, I concluded that the rebuttal successfully reinforced the soundness of their empirical claims.

**Key Questions For Authors:**

* Regarding the generation of semi-natural data in Experiment 2 (and the associated Algorithm 1), it would be very helpful if you could provide more details on how the "Reliability" parameter was applied. Specifically, I am curious whether the conditional probability $P(B|A)$ is manipulated uniformly across any A (i.e., applying the same reliability threshold regardless of the specific token $A$), or if the manipulation preserves the natural variance of the original distribution, where certain anchor tokens $A$ might inherently possess a much higher probability of being followed by $B$. Understanding this detail is important, as a strictly uniform application might mask the natural token-specific variance that could be helpful for bootstrapping induction heads.

**Limitations:**

yes

**Strengths And Weaknesses:**

### Strengths

* **Soundness:** Although the paper's primary contribution is rooted in empirical analysis, the breadth and rigor of its experimental validation are major strengths. The authors conduct experiments across a highly diverse set of datasets. They do not merely rely on natural language data (CC100), but meticulously construct semi-synthetic data to control specific bigram properties, and further employ fully synthetic data—completely decoupled from natural language—to broadly manipulate characteristics such as marginal distributions.

* **Presentation:** While the paper is densely packed with three comprehensive experiments and a substantial amount of information, it remains highly readable. The specific intent behind each experimental setup is explicitly stated, and the resulting empirical findings are presented in a clear and accessible manner.

* **Significance:** The detailed investigation into the precise conditions required for induction head (IH) formation holds substantial importance for the field. From the perspective of LLM pretraining on actual natural language datasets, it is quite remarkable that the exact timing of this phase transition can be predicted using only a small set of parameters. Furthermore, by focusing on the presence of bigram repetitions within a context and the conditional probability upon repetition ($P(B|AB\dots A)$), the discovery that IH formation is governed by a Pareto frontier is a significant finding that is highly likely to influence future mechanistic interpretability research.

* **Originality:** Although this overlaps somewhat with its significance, the paper demonstrates strong originality in its methodological approach to exploring IH formation conditions. Specifically, the methodology of meticulously controlling the statistical information of the training data—such as isolating the occurrence of in-context bigram repetitions and manipulating their conditional probabilities—offers a highly novel and rigorous lens through which to study the emergence of in-context learning.


### Weaknesses

* **Soundness:** The experimental validation could benefit from a wider diversity of evaluated LLM configurations. While the authors identify batch size, context length, and model size as predictive factors for the timing of induction head (IH) emergence, there are other potential factors in model training that might also play a role. Variables such as the number of attention heads, number of layers, embedding dimensionality, the choice of positional embeddings, and the learning rate could potentially influence IH formation. Adding ablation studies or further discussion regarding these architectural and hyperparameter choices would help clarify the robustness and generalizability of the proposed predictive law.

* **Presentation:** While the overall manuscript structure is logical, there are a few areas where the presentation of experimental details could be improved. For instance, the specific context length used in Experiment 2 (Section 6) does not appear to be explicitly mentioned in the main text. Given that the probability of in-context repetitions is closely tied to context length, including this parameter in the main body would aid reader comprehension. Furthermore, the exact method for manipulating "Reliability" could be clarified. Reliability is conceptually tied to the conditional bigram probability, $P(B|A)$, which in natural language exhibits large variance depending on the specific token $A$. It is slightly unclear whether this reliability threshold is applied uniformly across all tokens or if the natural variance is preserved. If applied uniformly, there is a possibility that it might inadvertently suppress high-probability anchor tokens that could naturally trigger IH formation. Clarifying how this token-dependent distribution is handled would strengthen the methodology section.

* **Significance:** The questions raised above regarding architectural diversity and the precise implementation of the bigram probability $P(B|A)$ suggest that the paper's specific quantitative findings might be somewhat sensitive to the chosen experimental setup. For instance, the exact numerical thresholds proposed (e.g., the observation that IHs fail to form at around 10% reliability) might potentially shift under different architectural configurations or if the training data retained more of its natural token-specific variance. While the qualitative findings are highly intriguing, discussing these potential dependencies would provide a more nuanced view of the results' broader applicability.

* **Originality:** The originality of the paper is solid. While the underlying metrics (such as the prefix-matching score) and the concept of an IH phase transition are established in prior literature, the core novelty of this work lies in its detailed manipulation of training data statistics—specifically, isolating in-context bigram repetitions and controlling their conditional probabilities. This explicit exploration of how specific data distributions govern the emergence of IHs provides a meaningful and original contribution to the field of mechanistic interpretability.

---

> ### Author Rebuttal · Authors · 2026-03-30
>
> Thank you for taking the time to review our paper, and for your thoughtful feedback.
>
> 1. **Other variables/training configurations**
> * We agree that a more comprehensive coverage of variables and training configurations would be ideal, and make the results more generalizable and robust. However, we would like to provide a few reasons why we limited our scope to the variables examined in this paper. First, given that almost all variables interact with each other, the combinatorics of our search space quickly explode, making it implausible to do more given our compute budget. Second, some of the variables have been studied in the previous literature; e.g. number of layers and heads have been widely studied as a prerequisite for IH emergence. Lastly, on top of our main GPT2 models (2 layers, 8 heads, ~ 50M), we have included larger GPT2s (up to 350M, trained from scratch), as well as Pythia variants for inference (up to 7B), which have different dimensions, numbers of layers, and numbers of attention heads. Yet the predictive law seemed to predict the emergence points quite well (cross-fold validations are included in Appendix D (p.12-)). We also show that the model size is not a significant predictor of emergence points, so we are fairly confident that the number of layers beyond $\ge$2, heads, and dimensionality do not play a significant role here.
> 2. **Context Length for Experiment 2**
> * We used the context size of 64, and we will include this in the main body of text - thank you!
> 3. **Generation constrained on frequency and reliability**
> * The full algorithm is described in Algorithm 1 (p.17); here, we touch on the parts relevant to the question. We applied the constraints of $P(AB…A)$ and $P(B|AB…A)$ uniformly at each dataset generation step, regardless of the token. However, because each token’s unigram and bigram probabilities are uniquely specified in the $|V| \\times |V|$ transition matrix, each token does have a different unigram and bigram distribution.
> * Now, we think that the uniform application of the contraints $P(AB…A)$ and $P(B|AB…A)$ does not mean each token loses its token-specific variance. At the first occurrence of A, we sample the next token based on A’s transition probability (A’s row from our transition matrix). At the second occurrence of A (i.e. at the AB…A pattern), we first decide (based on the reliability constraint) if the next token should be B or a different token. If yes, we will sample B, otherwise, we will sample based on the transition probabilities (minus B) from our transition matrix. We agree that a non-uniform application of the constraints (i.e. constraining the aggregate frequency and reliability while varying them from token to token) might be interesting, and it comes down to the question of to what extent each word in natural language differs in its probability of participating in repeated bigrams **beyond** their differences in unigram and bigram distributions. We will leave this to future work - we think our simple approach should suffice for the purpose of this study.
>
> 4. **Limitation of the current approach**
> * We agree that changing other variables training configurations could potentially change the results (although for many of them, we think the results are quite robust, as outlined in #1). In particular, as you pointed out, the implementation of frequency-reliability-constrained data generation could change the exact values we observed in Experiment 2. We would like to point out, however, that the frequency-reliability threshold found in the natural experiment (Experiment 1) and in the semi-natural experiment (Experiment 2) largely agree with each other, suggesting the validity of our methods in Experiment 2.
> * More specifically, in Experiment 1, when we experimented with various context sizes, context sizes of 4, 8, and 16 didn’t lead to the emergence of IHs, from which we can infer that the amount of bigram repetitions in 16 (fail) and 32 (success) is at the decision boundary (of whether or not IHs form). Upon examination, as shown in Figure 8 in Appendix F (p. 16), naturally sampled texts with context size 32 have frequency and reliability of 11.6% and 6.7%, respectively.
> * Now, back to Experiment 2, because frequency and reliability constraints are imposed on the second half of each chunk (to make sure tokens can vary freely in the first half - otherwise we find the algorithm producing degenerate sequences with high frequency and reliability constraints), the actual values for frequency and reliability are half of the imposed values. With this in mind, the 11.6% frequency threshold found in the natural experiments corresponds to somewhere bet ween 10% and 30% rows of Figure 4, and 6.7% reliability threshold roughly corresponds to the 10% column, which is exactly at the decision boundary. With these values obtained from natural and semi-natural experiments agreeing with each other, we think that the implementation suffices to produce robust results.

---

> > ### Author Rebuttal · Reviewer_bmrH · 2026-04-02
> >
> > Thank you for your detailed and thoughtful rebuttal.
> > Regarding my initial question about the uniform manipulation of the reliability parameter $P(B|A)$, I was mainly wondering if this approach might mask the natural token-specific variance, making the quantitative findings (like the ~10% threshold) difficult to interpret for natural language.
> > I really appreciate your clear explanation, and I agree that exploring non-uniform constraints is a great direction for future work. More importantly, your explanation showing how the empirically observed thresholds in the natural data (Experiment 1) interestingly align with the effective thresholds in the semi-natural data (Experiment 2, accounting for the half-sequence constraint) was very helpful. This resolved my initial concern to some extent and demonstrated the validity of your findings. Additionally, the extra context regarding the Pythia 7B model is a good point that reinforces the soundness of the predictive law.
> > In light of this convincing rebuttal, I am happy to raise my score from Weak Accept to Accept.
> >
> > As a minor suggestion, since future readers might naturally have the exact same question about the uniform application, it would be wonderful if you could briefly include what you explanted here in the Limitations or Future Work section, space permitting.

---

> > > ### Author Response · Authors · 2026-04-02
> > >
> > > Thank you for updating your evaluation on our paper - we are glad that our clarification helped address your concerns. We agree that the points raised in this discussion are of interest to other readers as well, and we will be sure to include them in the final draft. Thank you again!

---

### Official Review · Reviewer_VHNY · 2026-03-13

**Soundness:** 3
**Presentation:** 4
**Significance:** 3
**Originality:** 2
**Overall Recommendation:** 5
**Confidence:** 3

**Summary:**

This paper studies an important problem regarding the development of induction heads (IHs) in LLM pre-training, especially on the kinds of training data that enable IHs to occur. Specifically, it explores the effect of batch size, context size, and bigram repetitions on the occurrence of IHs. Experiments are done in both natural language and synthetic data settings.

**Compliance With Llm Reviewing Policy:**

Affirmed.

**Final Justification:**

My concerns have been adequately addressed and I am keeping my positive evaluation.

**Key Questions For Authors:**

1. In Figure 2, why are the best PS values on the rightmost plot much lower compared to the best PS values on the other plots?

2. Could you provide some intuitions on why reliability is more important than frequency for the emergence of IHs?

3. What are some practical implications of this work in terms of the construction of training data for LLM training?

**Limitations:**

Yes

**Strengths And Weaknesses:**

**Strenghts:** This paper is technically sound. The methods are well-described and empirical studies are rigorously performed to justify their theories. I particularly like the synthetic data example where controlled perturbations of the datasets are done in order to study the problem in more detail.

**Presentation:** This paper is very well-written and easy to follow.

**Significance:** This paper addresses an important problem on the emergence of induction heads in LLM pre-training.

**Originality:** This study utilizes existing techniques (e.g., prefix scores, scaling law) applied to understanding the pre-training datasets that allow for the development of induction heads.

---

> ### Author Rebuttal · Authors · 2026-03-31
>
> Thank you for taking the time to review our paper, and providing thoughtful comments.
>
> 1. > In Figure 2, why are the best PS values on the rightmost plot much lower compared to the best PS values on the other plots?
> * Here, we forced $X\\%$ of training chunks (chunk=$k$ tokens, where $k$ is the context length)  to have no bigram repetition at all. With the context size of 64 (which is the configuration used in the rightmost plot of Figure 2), in natural training example, almost all chunks have at least 1 bigram repetitions, promoting the higher best PS. Because we are forcing higher proportions of chunks with **no** bigram repetition at all (30%-70%), these data suppress the strength of the induction heads, hence lower best PS scores. When 70% of the chunks have no bigram repetitions at all, induction heads do not seem to emerge.
> 2. > Could you provide some intuitions on why reliability is more important than frequency for the emergence of IHs?
> * Intuitively, frequency P(AB…A) could be thought of as the number of tokens where an inductive completion (AB…AB) is possible, and could be potentially rewarded. On the other hand, reliability P(B|AB…A) is how many of these potential opportunities are actually rewarding the inductive completion behavior. In that sense, even if there are many tokens that match the AB…A pattern, if predicting B is not reinforced as the correct prediction, there is no incentive for the model to learn this behavior. The conceptual explanations of these two values are provided in appendix F (p.15-), but we did not explain why the model is more sensitive to reliability. We will add this explanation to the text - thank you!
> * We also received a question about the functional form of the Pareto Frontier, and we did a follow-up analysis: We first directly model the PS for each $P(AB…A), P(B|AB…A)$ combinations: $\\hat{PS}=\\sigma(k(\\alpha\\log p_a+\\beta\\log p_b-\\tau))$, where $p_a$ and $p_b$ are $P(AB…A), P(B|AB…A)$, respectively. Fitting this to our data, we obtain $\\alpha=0.5674, \\beta=1.1531, \\tau = -2.6407, k = 9.5950$. Here, we can see that $\\beta$ is more than twice as large as $\\alpha$, matching our descriptive observation that the IH formation is more sensitive to reliability than to frequency.
>
> 3. > What are some practical implications of this work in terms of the construction of training data for LLM training?
> * We believe the main contribution of this paper is more about understanding the training dynamics and data distributional properties de/promoting the emergence of induction heads. However, as to the practical implications for dataset construction, we suspect that tokenization plays a significant role in deciding the number of repetitions in the text. Subword tokenization artificially increases the number of repetitions (if 1 relatively long word appears in context twice, this is considered AB…AB; e.g., “encoders and decoders” → [enc, **od, ers,** and, dec, **od, ers**]). If a dataset is constructed with tokenizers that use larger units (words, multi-word tokens, etc.), this could potentially weaken the emergence of induction heads.

---

> > ### Author Rebuttal · Reviewer_VHNY · 2026-04-03
> >
> > Thank you for your comments. I am keeping my positive score.

---

### Decision · Program_Chairs · 2026-04-30

**Decision:**

Accept (regular)

**Comment:**

This paper studies the emergence of induction heads during language-model pretraining and characterizes how this depends on training configuration and data statistics. The submission makes three main contributions: a predictive law for emergence timing, an analysis of bigram repetition frequency and reliability, and synthetic-data experiments that further dissect the conditions under which induction heads form. Reviewers were generally positive about the clarity, rigor, and breadth of the empirical evidence, though some still raised generalization and presentation concerns.

The paper's main strengths are its careful experimental design and its ability to move beyond anecdotal observations. In particular, the predictive law for emergence timing and the decomposition of burstiness into more interpretable statistical factors are well recognized among the reviewers. The main reservations concerned novelty scope and how broadly some conclusions generalize, but these concerns were relatively modest compared with the paper's overall strengths.

The rebuttal was well received and further increased confidence. Overall, I recommend accept. For the final version, the authors should continue to clarify novelty relative to closely related prior work and incorporate the additional revision, discussion and analyses promised during rebuttal.